# Particle size dynamics in abrading pebble populations

András A. Sipos[1,2], Gábor Domokos[1,2], and János Török[1,3]

[1] MTA-BME Morphodynamics Research Group Budapest University of Technology and Economics, Műegyetem rakpart 1-3, Budapest, Hungary
[2] Department of Materials and Structures, Budapest University of Technology and Economics, Műegyetem rakpart 1-3, Budapest, Hungary
[3] Department of Theoretical Physics, Budapest University of Technology and Economics, Budafoki út 8, Budapest, Hungary

**Correspondence:** András A. Sipos (siposa@eik.bme.hu)

**Abstract.** Abrasion of sedimentary particles in fluvial and aeolian environments is widely associated with collisions encountered by the particle. Although the physics of abrasion is complex, purely geometric models recover the course of mass and shape evolution of individual particles in low and middle energy environments (in the absence of fragmentation) remarkably well. In this paper, we introduce the first model for the collision-driven *collective* mass evolution of sedimentary particles. The model utilizes results of the individual, geometric abrasion theory as a *collision kernel*, following techniques adopted in the statistical theory of coagulation and fragmentation, the corresponding Fokker-Planck equation is derived. Our model uncovers a startling fundamental feature of collective particle size dynamics: collisional abrasion may, depending on the energy level, either focus size distributions, thus enhancing the effects of size selective transport or it may act in the opposite direction by dispersing the distribution.

# 1 Introduction

## 1.1 Geological observations

Probably the most fundamental observation on pebbles is that they appear to be segregated both by size and shape and it is broadly accepted that the dynamics are driven by two physical processes: transport and abrasion. Which of these processes dominates may depend on the geological location and also on timescales, however, geologists appear to agree that, in general, neither process should be ignored.

In *coastal environments*, one of the most remarkable accounts of pebble size and shape distribution is provided by Carr (1969) based on the measurement of approximately a hundred thousand pebbles on Chesil Beach, Dorset, England. In summarizing his results, Carr provides mean values and sample variations for maximal pebble size and pebble axis ratios along lines orthogonal to the beach. These plots reveal pronounced segregation by maximal size and shape, i.e. on shingle beaches pebbles of roughly similar maximal sizes and with roughly similar axis ratios appear to be spatially close to each other. Size and shape segregation has been broadly observed in various settings (Bird, 1996; Gleason et al., 1975; Hansom and Moore, 1981; Kuenen and Migliorini, 1950; Neate, 1967) and it was mostly attributed to the global transport of pebbles by waves (Lewis, 1931; Carr, 1969) but, in some settings, may also be related to abrasion. Indeed, a detailed account of the interaction of abrasion and transport is given by Landon (1930) who investigated the beaches on the west shore of Lake Michigan. He attributes size and shape variation to a mixture of abrasion and transport. Kuenen (1964) discusses Landon's observations, however disagrees with the conclusions and attributes size and shape variation primarily to transport. Carr Carr (1969) observes dominant sizes and shape ratios emerging as a result of abrasion and size grading while Bluck (1967) describes beaches in South Wales where equilibrium distributions of size and shape are reached primarily by transport and abrasion plays a minor role. Which of the two processes (transport or abrasion) dominate may well depend on the timescales they operate on. While abrasion appears in some scenarios to act much slower than transport, a recent study (Bertoni et al., 2016) verified mass losses on the order of 50% on a pebble beach over a 13-months period, indicating that in some settings the two processes may indeed compete in determining size and shape distributions.

In *fluvial environments*, while downstream fining of sediment has been often attributed to transport Paola et al. (1992); Ferguson et al. (1996); Fedele and Paola (2007); Whittaker et al. (2011), other authors have pointed to the significance of attrition Brewer and Lewin (1993); Attal and Lavé (2006); Dingle et al. (2017). In Miller et al. (2014) the authors, using field data, provide quantitative assessment of the significance selective transport with respect to attrition in downstream fining. Beyond the evolution of smooth size and shape distributions, there is yet another common phenomenon in fluvial geomorphology where the interaction of transport and attrition could be far from trivial. The often observed presence of isolated large boulders in rivers Huber et al. (2020) may be explained solely by transport, as these large pieces are often not carried by the river, rather, they move by a different process (e.g. landslide or debris flow). On the other hand, these large rocks could also be interpreted as *outliers* emerging spontaneously in a pebble size distribution on which collisional abrasion certainly has strong impact in upper reaches of rivers.

As we can see, both in coastal and fluvial environments it is a generally accepted fact that the two processes (transport an attrition) appear to *compete* in shaping the evolution of pebble shape and pebble mass distributions. How exactly this competition may play out and in what manner attrition may contribute to this process is the subject of our paper.

We also remark that while all available observations indicate that attrition could be a relevant factor in the evolution of shape and mass distributions, so far, in the absence of any predictive theory, no datasets have been collected which would admit to verify any theoretical predictions. We will point out potential strategies for verification in Section 4.

## 1.2 Existing theory

### 1.2.1 Individual abrasion

*Individual abrasion* is a theory describing the mass and shape evolution of one individual particle (*abraded particle*) under the impacts of many incoming particles (*abraders*) (see Figure 1(a)). In the mean field theory for the geometry of individual abrasion only the mass and shape of the abraded particle is recorded, the effect of impacts is averaged and the evolution is determined by the size of the abraded particle compared to the average size of the abrading particles.

The mean field geometric theory of individual abrasion (i.e. shape evolution) for sedimentary particles under collisions is, since the seminal papers by Firey (1974) and Bloore (1977), well understood and validated (Szabó et al., 2013; Szabó et al., 2015; Novák-Szabó et al., 2018). Still, despite the success of the Firey-Bloore geometric theory of shape evolution it was clear (Domokos and Gibbons, 2018) that it is not suited to predict the evolution of size: in stark contrast with geological observations summarized in Sternberg's Law (Sternberg, 1875), predicting exponential decay of particle mass and *infinite lifetime* for all particles, geometric abrasion theory predicted *finite lifetime* for all particles. On the other hand, Sternberg's broadly accepted theory of mass evolution (Sternberg, 1875) had nothing to offer regarding the evolution of shape. Recognizing this challenge, in (Domokos and Gibbons, 2018) a unified theory, called *volume weighted shape evolution* has been proposed which, on one hand, reproduces all the geometric features of the Firey-Bloore geometric theory, on the other hand, it also predicts mass evolution in accordance with Sternberg's Law.

### 1.2.2 Binary abrasion

The first stepping stone between the theory of individual abrasion and collective abrasion is the model for *mutual, binary* abrasion where two particles mutually abrade each other and we track both evolutions (see Figure 1(b)). In this case one can still write mean field equations by averaging over many collisions and the mass and shape evolution of both particles are recorded. For any binary abrasion model of size evolution, we postulate the following requirements:

 – size evolution should follow Sternberg's Law,

 – mass loss in a collision should be a monotonically increasing function of collision energy and

 – the model should be fully compatible with the geometric evolution model.

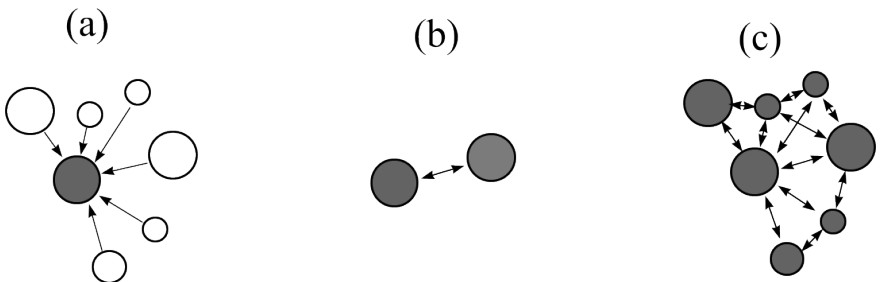

**Figure 1.** Schemes for (a) individual abrasion (Firey, 1974; Bloore, 1977), (b) binary abrasion (Domokos and Gibbons, 2018) and (c) collective abrasion, respectively. Volume loss only tracked for shaded particles. Arrows represent (non-simultaneous) collision events between particles.

The unified theory in (Domokos and Gibbons, 2018) offers a model satisfying all three requirements: by extending the Firey-Bloore equations and Sternberg's theory and using the kinetic energy of collision, models for *binary shape evolution* and for *binary mass evolution* of two mutually abrading particles were put forward. These two models have been merged in (Domokos and Gibbons, 2018) into a unified volume weighted theory of binary abrasion, compatible both with the Firey-Bloore and with
the Sternberg theory. The volume weighted model for binary mass evolution, describing the time evolutions for the masses $X(t), Y(t)$ of two particles with respective material properties $m_1, m_2$ can be written as

$$X_t = -c_{12}\frac{XY}{X+Y}, \tag{1}$$
$$Y_t = -c_{21}\frac{YX}{X+Y}, \tag{2}$$

where the subscript $t$ refers to differentiation with respect to time and the constant prefactors $c_{12}$ and $c_{21}$, which we call the
*binary abrasion parameters*, depend simultaneously on the materials $m_1$ and $m_2$ of the $X$ and $Y$-particles, respectively.

We also note that in case of two *identical* particles (e.g. two particles with identical masses $X = Y$ and identical material properties $c_{XY} = c_{YX}$) the system (1) and (2) predicts mass evolution according to Sternberg's Law. In case of different masses or properties we still have infinite lifetime with one of the particles approaching zero mass asymptotically as times goes to infinity and the other particle approaching a finite mass.

**1.2.3   Collective size dynamics**

Independently of individual (and binary) abrasion theory there exists broad interest in *collective* shape and size evolution models tracking mutually colliding populations of $N$ particles (see Figure 1(c)). Similar problems arise in particular in the context of coagulation (da Costa, 2015) and dynamic fragmentation processes (Cheng and Redner, 1988). In such collective evolution models the main question is how the size distribution of particles, starting from an initial distribution, evolves in time
due to the mutual collisions. These models use a standard framework relying on a so-called *collision kernel*. In a more general

setting, the collision kernel is referred to as the *interaction kernel*. Our choice of terminology is motivated by the fact that in our case the only interactions are collisions.

The collision kernel can be derived from the binary equations (the physical model of the $N = 2$ case) by incorporating statistical effects, i.e. that collision probability may depend on particle speed or mass. In (Domokos and Gibbons, 2018) the binary model (1) and (2) was extended to a kernel by introducing an additional scalar parameter $r$ (to which we will also refer to as the *environmental parameter* of the evolution), representing the assumption that on the average, only the collision probability depends on particle size and the collision speed is independent of mass:

$$X_t = -c_{12} \frac{X^{1+r} Y^{1-r}}{X + Y}, \tag{3}$$

$$Y_t = -c_{21} \frac{Y^{1+r} X^{1-r}}{X + Y}. \tag{4}$$

Note that these equations are identical to the formulas (118) and (119) in (Domokos and Gibbons, 2018) with $\alpha = 0$ in their notation and taking $r = \nu$, $X = V_X$, $Y = V_Y$, $c_{12} = c_{21} = c$. Alternative interpretations of $r$ are also possible; we will discuss the role of the environmental parameter $r$ in detail in Subsection 2.4. Henceforth, in the main body of this paper (apart from Appendix A) we assume that the pebble population is *homogeneous*, i.e. that the material for all pebbles is identical so we have $c = c_{12} = c_{21}$ and the sole role of the constant $c$ is to set the timescales. We will incorporate this into the time variable $t$ and henceforth, for homogeneous pebble populations, we set $c \equiv 1$. We will discuss the role and identification of material constants in heterogeneous pebble populations in Appendix A.

Once the kernel has been established, we make the assumption that for large $N$ the collective size evolution is a stochastic process driven by many binary events among the particles, implying that the core of the collective process is still the above mentioned collision kernel. This allows for the construction of the *master equation*, also known as the Fokker-Planck equation which describes the time evolution of the particle size distribution. Although the collective abrasion is a stochastic process, in the $N \to \infty$ limit the collision kernel will uniquely determine the global evolution of the continuous size distribution. The master equation (or Fokker-Planck equation) is expressing this evolution. Determining the master equation based on the collision kernel is the second step in the statistical model.

## 1.3 Our model

### 1.3.1 Relationship to earlier models

The above-outlined structure is characteristic for *coagulation fragmentation* models (da Costa, 2015), in particular for non-linear fragmentation, which describe fragmentation processes triggered by binary collisions of particles. Our model may be regarded as a special case of the non-linear fragmentation models (Cheng and Redner, 1988) since, in addition to the standard framework adopted in these models we also make two simplifying assumptions:

1. we only consider collisions where the relative mass loss is small (i.e. the particles lose only fragments with small relative mass) and

2. the small fragments generated in the collisions are not considered further in the evolution.

By implementing these two assumptions into the statistical model based on the collision kernel (3) and (4), we take the first step towards establishing the statistical theory of collective size and shape evolution of sedimentary particles. This approach offers multiple methodological advantages. On one hand, by using (4) as the collision kernel, our statistical model will be compatible with Sternberg's Law so we can expect the collective evolution also to observe this theory, albeit in a statistical sense. On the other hand, we can also expect all our results to be compatible with an extended (future) theory which also describes collective shape evolution based on the unified, volume weighted geometric theory in (Domokos and Gibbons, 2018).

### 1.3.2 Basic notations

To describe our construction we will need to address both the size evolution of individual particles (under the collision kernel) as well as the evolution of size distributions. While particle size appears in both settings, we need to distinguish carefully: in individual and binary models particle size evolves in time, in collective models size distribution evolves in time. As a consequence, in the individual setting the variable denoting size may be differentiated with respect of time, in the collective setting this is not the case. We will use $X, Y$ to denote individual particle sizes (either volume or mass) and we will use $x, y$ to denote the independent variables of size distributions. Time evolution of individual particle size will be denoted by $X(t), Y(t)$ with time derivatives $X_t(t), Y_t(t)$ (arguments of a function written in subscript will refer to differentiation throughout the paper). The time evolution of size densities will be denoted by $f(x,t), f(y,t)$ with time derivatives $f_t(x,t), f_t(y,t)$ and size-derivatives $f_x(x,t), f_y(y,t)$. We denote the expected value and variance of these size distributions respectively by $E(t)$ and $W(t)$ and we will primarily use the relative variance $R(t) = W(t)/E(t)^2$ to characterize the evolution of the distributions.

### 1.3.3 Main results

The collision kernel (3) and (4) for mass evolution in (Domokos and Gibbons, 2018) has one single environmental parameter $r$ which is inherited by the corresponding Fokker-Planck equation (shown in Subsection 2.2). As we will describe in Section 2, the environmental parameter $r$ may, depending on interpretation, represent either the size dependence of the number of collisions or, alternatively, the size dependence of collision energy. Regardless of the interpretation, in Subsection 2.3 we find that the value $r = 0.5$ is *critical* as it separates two regimes of collective abrasion with qualitatively different evolution $R(t)$ of the relative variance :

– For $r > 0.5$ we find *focusing* processes with decreasing $R(t)$, approaching $R(t) = 0$ in the limit as time approaches infinity. Here the size distribution converges to a Dirac delta function. This parameter range corresponds to lower energy levels. Natural abrasion processes belonging to this regime will thus amplify the segregating effects of size-selective transport.

– For $r < 0.5$ we find *dispersing* processes with increasing $R(t)$, thus counter-acting size-selective transport processes. This corresponds to collisional abrasion at higher energy levels.

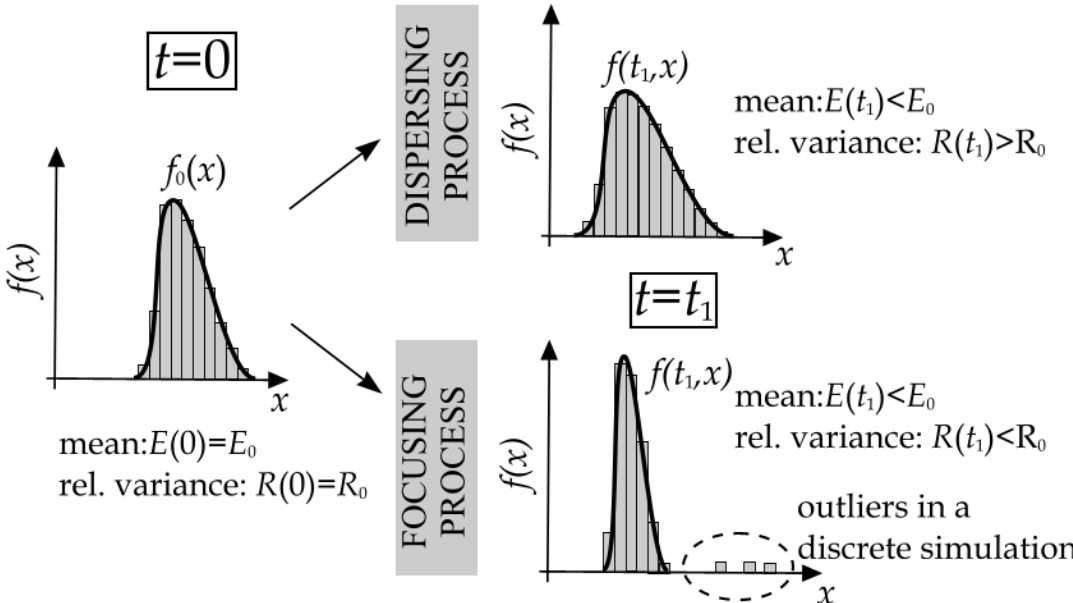

**Figure 2.** Schematic description of the evolution of mass distribution of a pebble population: in a dispersing process the relative size variation $R(t)$ of the a mass distribution, either represented by an empirical histogram (Carr, 1969), or a continuous function ($f_0(x)$) at $t = 0$, is increasing. In the continuum model of a focusing process $R(t)$ decreases as time evolves however, in a discrete model with a finite number of particles some outliers appear (indicated by dashed ellipse) with mass substantially above the average. The reduced distribution (without the outliers) produces a decreasing relative variance, analogous to the continuous model.

As collisional abrasion may occur at a broad range of energies, these two basic scenarios of the model (illustrated in Figure 2) offer an explanation for the broad range of geological observations (Bluck, 1967; Landon, 1930; Carr, 1969; Kuenen, 1964) in relating the relative significance of transport and abrasion in various scenarios. Our model also reflects the universality of Sternberg's Law by predicting, regardless of the environmental parameter $r$, exponential decay as the universal evolution $E(t)$ of the expected value.

In general, the evolution equations generated by (3) and (4) for the mean $E(t)$ and the variance $W(t)$ are integro-differential equations which are hard to solve analytically. To support our claims, we will use three types of approximations:

(a) We approximate the kernel (3)-(4) by its truncated Taylor series expansion and investigate the evolution of general initial density functions. This is found in Appendix C1.

(b) We regard the full kernel, however, we only investigate density function obtained as a small perturbation of the Dirac delta (i.e. populations of *almost* identical particles). This is done in Appendices C2 and C3.

(c) We numerically compute both the discrete and the continuum models. For details see Section 3.

We will briefly refer to the first two approximations as the *continuum model*. In case of the third approximation we do direct, discrete simulations of finite particle populations we use the full kernel and we call this the *discrete model*. One startling feature of the latter (as compared with the former) is the appearance of *outliers*, i.e. particles substantially larger than the vast majority (illustrated in Figure 2). As we can observe, the bulk of the density function closely mimics the evolution in the continuum model. The quantitative analogy in the evolution of the relative variation $R$ can be also recovered if we consider a reduced density function $f^\star(t_1, x)$ by omitting the outliers, i.e by applying an upper cutoff in size, omitting bins containing only one particle. The reduced density function $f^\star(t_1, x)$ is characterized by the reduced relative variation $R^\star$ which will decrease in a focusing process, however, in contrast to the continuum model, it will not approach zero, rather a positive constant.

## 1.4 Testing of the model for homogeneous pebble populations

As outlined above, our model is defined on two levels: the collision kernel (3)-(4) we will briefly refer to as the *input level* as it defines the basic physics of the underlying collisions. The Fokker-Planck equation we will briefly refer to as the *output level* as it defines the evolution of the mass density function based on the collision kernel. One may test the model at both levels. Below we discuss the case of homogeneous pebble populations where the evolution of the mass distribution is controlled by the single material parameter $c$ and the single environmental parameter $r$:

(a) One may test the model at the *input level*, by fitting the kernel (3)-(4) to laboratory tests where abrasion rate is plotted as a function of particle size. Such an experiment could be used to determine the material parameter $c$ for a given homogeneous population. Also, if the laboratory test is imitating the environment of the natural process, the environmental parameter $r$ may be also obtained in this manner. We also note that the functional relationship between particle size and abrasion rate will not only depend on the parameters but also on particle size. For details, see Appendix A.

(b) One may test the model at the *output level* by measuring the time evolution of full mass distributions and fitting the respective material and environmental parameters $c$ and $r$ to this dataset. While we are not aware of any such public dataset, this could be performed in a laboratory either in a flume or in a drum experiment. In the field the optimal solution appear to be radio-tagged pebbles (Bertoni et al., 2016).

The above simple procedures apply only for homogeneous populations. We lay out the procedures for the testing of the model for heterogeneous populations in Appendix A where we also perform partial testing for the laboratory data obtained by (Attal and Lavé, 2009).

## 2 Modeling collective size dynamics

### 2.1 General form of the collision kernel

The first simplification described in Subsection 1.3.1 implies that the limit where relative fragment mass approaches zero offers a good approximation, thus it admits a collision kernel of the type used in Ernst and Pagonabarraga (2007), describing

continuous mass evolution via coupled ordinary differential equations for the evolution of particles with masses $X(t)$ and $Y(t)$:

$$-X_t = \psi^1(X,Y), \tag{5}$$

$$-Y_t = \psi^2(X,Y), \tag{6}$$

where $\psi^1(X,Y)$ and $\psi^2(X,Y)$ are differentiable ($C^1$) functions, with positive values (i.e. $\mathbb{R}^+ \times \mathbb{R}^+ \to \mathbb{R}^+$). Symmetry of
the binary process implies $\psi^1(X,Y) = \psi^2(Y,X)$, so often superscripts are suppressed and the kernel is simply referred to as
$\psi(.,.)$. Selection of the kernel encapsulates not only the physics of binary collisions, it also may include the mass-dependent
probability of collision between two particles. We will discuss the identification of a physically sound kernels in Subsection 2.3.

## 2.2  General form of the master equation

The second simplification in Subsection 1.3.1 admits the construction of the master equation solely based on the collision
kernel (by omitting additional terms for the remainder of the fragmented material). These simplifying assumptions also set
our model apart from general fragmentation models in another respect: in the latter, constant mass is prescribed as a global
time-invariant while the (integer) number of particles changes whereas in our model total mass is decreasing while the number
of particles remains constant and serves as a global invariant.

Using these considerations, for our problem the master equation is found to be

$$f_t(t,x) = \frac{\partial}{\partial x}\left[ f(t,x) \int_0^\infty f(t,y)\psi(x,y)dy \right] = f_x(t,x) \int_0^\infty f(t,y)\psi(x,y)dy + f(t,x) \int_0^\infty f(t,y)\psi_x(x,y)dy, \tag{7}$$

where subscripts stand for partial derivatives. Without loss of generality, the evolution starts at $t = 0$ and we consider the initial
distribution of the volume $f(0,x) \equiv f_0(x)$ to be *a-priori* known. Note that contrary to the majority of Fokker-Planck models,
our model contains solely the advection term, which readily follows from the deterministic nature of the kernel. Here we aim
to determine the collective behavior implied by (7). Nonetheless, a stochastic kernel would produce diffusion in the master
equation. Such a generalization would inevitably reduce the analytic transparency and thus the qualitative predictive capability
of the model. Whether or not it is justified from the quantitative point of view, can be decided based on extensive testing
campaigns.

We aim to understand some scenarios characteristic of pebble populations by investigating the Cauchy-type initial value
problem associated with equation (7), starting at the distribution $f_0$ with mean value $E_0$, variance $W_0$ and relative variance
$R_0 := W_0/E_0^2$.

## 2.3  Collision kernels

Detailed physical modeling of the collisional event can make the interaction kernel highly complex; for a recent review on
kernels see (Meyer and Deglon, 2011). On the other hand, mathematical studies tend to prefer simple expressions for $\psi(X,Y)$,
admitting rigorous, analytical conclusions. Our goal is to a find kernel which has strong physical basis, yet it admits an analyt-
ical approach, thus it offers a trade-off between between physical and mathematical preferences.

We first consider two simple kernels which satisfy the mathematical requirement of leading to analytically soluble Fokker-Planck equations. However, as we will show, these very analytical results highlight that these kernels are physically not admissible. Next, we investigate the parameter dependent compound kernel suggested in (Domokos and Gibbons, 2018) which grabs the essential physics of the investigated process, yet, the corresponding Fokker-Planck equation still admits analytical conclusions.

First, we consider the *summation kernel* (denoted by $\{.\}^+$), where the mass loss rate is proportional to the sum of the masses of the colliding particles:

$$\psi^+(X,Y) := X + Y, \tag{8}$$

stating that the rate of mass loss in binary collisions is proportional to the total mass of the two colliding particles. Appendix B1 demonstrates that the relative variance of the mass in case of the summation kernel follows $R^+(t) = R_0 e^{2t}$, hence it is a *dispersive* process regardless of the initial distribution $f_0$.

In the very same manner let us investigate the *product kernel* distinguished by the sign $\{.\}^*$. The product kernel is defined via

$$\psi^*(X,Y) := XY. \tag{9}$$

According to Appendix B2, the relative variance in this case is constant as $R^*(t) = R_0$ for all $t \geq 0$, which that the model is nor focusing neither dispersing. Note that the time-invariance of $R^*(t)$ under the product kernel does not imply the invariance of the P.D.F $f(t,x)$ per-se. In addition, we see a polynomial decay in the mass as $E^*(t) = (t + E_0^{-1})^{-1}$, which contradicts Sternberg's law (Sternberg, 1875) that postulates an exponential decay.

In order to be in accordance with Sternberg's law and to have a control on the evolution of the relative variance, following the lead of (Domokos and Gibbons, 2018) we investigate the interaction law (3) and (4) which we call a *compound kernel* and using the introduced general notation for kernels, we distinguish it with the $\{.\}^c$ sign:

$$\psi^c(X,Y) := \frac{X^{1+r}Y^{1-r}}{X+Y}, \tag{10}$$

where $0 \leq r \leq 1$ is a fixed parameter. Henceforth we investigate evolution of mass density functions under the Fokker-Planck equation derived from (10). In Appendix C1 we show analytical results for the evolution if the kernel (10) is replaced by its truncated Taylor expansion. In Appendices C2 and C3 we show analytical results for the evolution under (10), using a Dirac delta as initial distribution. The evolution under (10) with no restrictions for the initial condition is studied numerically. The essential properties of the three investigated kernels are summarized in Figure 3.

## 2.4 Interpretation of the parameter $r$

In natural events, both velocity and collision probability (cross-section) may depend on particle size: in laminar flows relative velocity and collision probability is proportional to linear size, while in a turbulent flows velocity could be inversely proportional to linear size and collision probability could be proportional to projected area. In the collision kernel (10) both effects

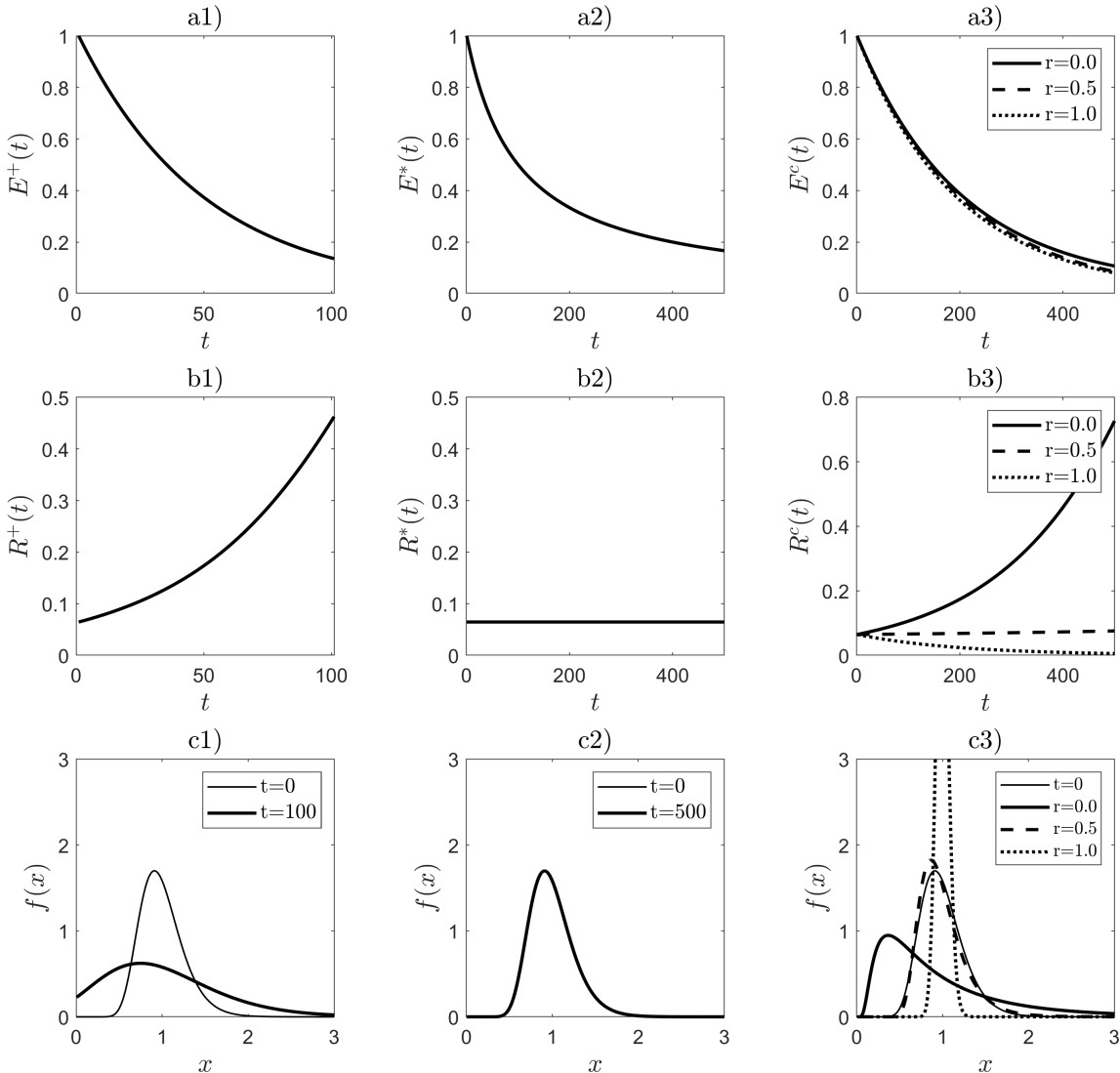

**Figure 3.** Evolution of an initial ($t = 0$) lognormal probability density function under Fokker-Planck equation generated by various kernels. Summation kernel: a1,b1,c1. Product kernel: a2,b2,c2. Compound kernel:a3,b3,c3. First row (a1,a2,a3): mean $E(t)$. Second row (b1,b2,b3): relative variance $R(t)$. Third row (c1,c2,c3): initial ($t = 0$) and final densities $f(x,t)$.

(dependence of velocity and dependence of collision probability on speed) are represented by the single scalar parameter $r$, so one may freely assign various interpretations to this parameter. In Domokos and Gibbons (2018) one particular interpretation was used: the compound kernel was derived using the assumption that particle velocity is independent of the size (e.g. rather determined by the surrounding fluid), but the collision probability goes as a power law with particle size, i.e. $X^r$. The effective mass combined with the collision probability gives the kernel in Eq. (10). However, alternative interpretations are possible,

the only essential underlying assumption is that we regard a one-parameter family of scenarios. In this family, if velocity is proportional to $X^a$ and collision probability is proportional to $X^b$ then we have $r \simeq a + b$.

To have a global view, it may be of interest to estimate the parameter $r$ in two extreme (limiting) scenarios. Laminar flows are characterized by a linear velocity profile. The particles hit each other if their trajectories intersect. Integration of the linear velocity profile combined with a spherical particle shape yields a collision probability proportional to $\sim X^{2/3}$ or, alternatively $r \simeq 2/3$. The other extreme case corresponds to turbulent flows, where we have *equipartition*, i.e. the kinetic energy of the particles is independent of their size (see e.g. Uberoi (1957)), implying that particle velocity is proportional to $X^{-1/2}$. Since the area of the cross-section is proportional to $X^{2/3}$ we arrive at a collision probability $X^{1/6}$, or, alternatively $r \simeq 1/6$. As we can see, both extreme scenarios yield $r$ values far away to either side of the critical value $r_{\mathrm{crit}} = 1/2$, so these estimates suggest that smooth steady conditions should result in a focusing and turbulent gas-like behavior in a dispersing process. For a detailed derivation see Appendix D.

In order to examine the validity of these assumptions we made discrete element simulations using the event driven method Lubachevsky (1991). In event driven dynamics, collisions are considered instantaneous and resolved accordingly which is best suited to obtain proper collision statistics. We emulated the above-mentioned processes by choosing an artificial mass for the particles and simulating a chaotic system. The artificial mass was used to obtain different volume-velocity relations in different scenarios. We found that in chaotic/turbulent systems that relative velocities were proportional to $v \sim X^{-1/2} - X^{-1/3}$ and the system behaved as the continuum model with $r \sim 1/6 - 1/3$. On the other hand, if velocities were proportional to $v \sim X^{-1/6} - X^0$ then the system was similar to a continuum model with $r = 1/2 - 2/3$. Thus the discrete element simulations fully support the results of the compound kernel.

## 2.5 Fluvial abrasion

Here we interpret the intuitive picture of fluvial abrasion in the context of our statistical model. In our model a fluvial environment may be represented by a *fluvial population*, consisting of $N + 1$ particles: a very large number ($N$) of small particles $X^i$ ($i = 1, 2, \ldots N$) representing the pebbles carried by the river and one very large particle $Y$ representing the riverbed. Such a scenario can not be explored directly in the context of our continuum model, however, as we will discuss in detail in Subsection 3.3, the discrete model can capture this situation even in the limit as $N \to \infty$.

To make a meaningful characterization of geologically relevant scenarios, we will regard two extreme cases which represent brackets on geological processes. In both cases we assume that the mass evolution is driven by binary collisions and we regard the limit as $N, Y \to \infty$ (while the masses $X^i$ of the small particles remain finite). Since we are interested in the mass evolution of pebbles (and in the current paper we are not interested in the mass evolution of the riverbed) we will denote the relative variance of the pebble population (i.e. all $X^i$ particles, the riverbed $Y$ not included) by $R(t)$. Our aim is to establish the sign of $R_t(t)$ as the main qualitative feature of collective dynamics, as $R_t(t) < 0$ and $R_t(t) > 0$ imply focusing and dispersing processes, respectively.

In the first extreme scenario we assume that particles are chosen uniformly from the full *fluvial population*: i.e., the riverbed has no special role. In this case *almost all* collisions will happen among a pair of small particles $(X^i, X^j)$ thus the presence of

the riverbed has no impact on the evolution of $R(t)$. For this extreme case all predictions of our continuum model remain valid: $r = 0.5$ will be a critical parameter value above which we see focusing ($R_t < 0$), below which we see dispersing ($R_t > 0$) behaviour. At the critical value $r = 0.5$ our model predicts neutral behaviour with $R_t = 0$.

In the second extreme scenario we assume that the small particles *exclusively* collide with the riverbed (large particle), i.e.,
we only have $(X^i, Y)$-type collisions. This means that the evolution for each of the small particles is an identical, independent two-particle process governed by the model (1)-(2) for binary collisional mass evolution. In this process, in the $Y \to \infty$ limit each individual small particle $X^i$ will thus evolve as

$$X^i(t) = X^i(0)e^{-t} \tag{11}$$

and thus follow Sternberg's Law. It is easy to show that for any initial distribution for the masses $X^i(0)$, in this process we
have $R_t = 0$. The large $Y$-particle (riverbed) will lose some mass as well but in this publication we are not interested in that part of the process.

Intuitively it is clear that any geologically relevant process is in-between the above two extreme cases and, although we do not deliver a rigorous proof, it appears plausible that in a geologically relevant setting $R_t$ will be also bounded by the two evolutions predicted for the two extreme scenarios. As for the second extreme scenario we have $R_t = 0$ we expect that for
any intermediate scenario the sign of $R_t$ will agree with the sign of $R_t$ based on the first extreme scenario. Our results show that the focusing behavior of the particle size distribution, or lack thereof, depends on interparticle interactions and not on the collisions between the particles and the river bed. This would imply that all our qualitative predictions remain valid in fluvial environments.

## 3   Numerical results

Here we perform computations to illustrate the main results presented in Subsection 1.3.3 by discretizing time with a fixed time-step $\Delta t$. The discrete model has been simulated with custom-made codes in Matlab and Python performing $M * [N/2]$ collisions between pairs during one time-step $\Delta t$, where $M$ is fixed model parameter and $N$ is the size of the population. The simulation starts with the creation of $N$ particles whose volumes are randomly sampled from the initial distribution $f_0$. Binary collisions are performed on uniformly selected pairs, i.e., all particles have equal chance of being selected irrespective of their
volume. Once a pair is selected, the collision kernel $\psi^c$ is applied and volume decrement is computed with time-step $\Delta t/M$. After the binary collision event both particles with reduced volume are replaced into the sample. In the presented simulations we set the population size to be $N = 5000$, the time-step $\Delta t = 0.01$ and $M = 10$. In the continuum setting $f(t,x)$ evolves under eq. (7) with some initial value $f_0(x)$. This code uses the operator exponential syntax of a the Chebfun toolbox (Driscoll et al., 2014) in Matlab.

## 3.1 Focusing and dispersing regimes

The evolution of a pebble population under the compound kernel was simulated both in the frame of discrete and the continuum model, i.e. by direct event-base simulation and by discretizing the partial differential equation. (see Sec. 1.3.3). The results show excellent agreement with our analytical predictions: $r = 1/2$ appears indeed a critical parameter in the model. This is illustrated in Figure 4 where a lognormal distribution is used as an initial value for the evolution.

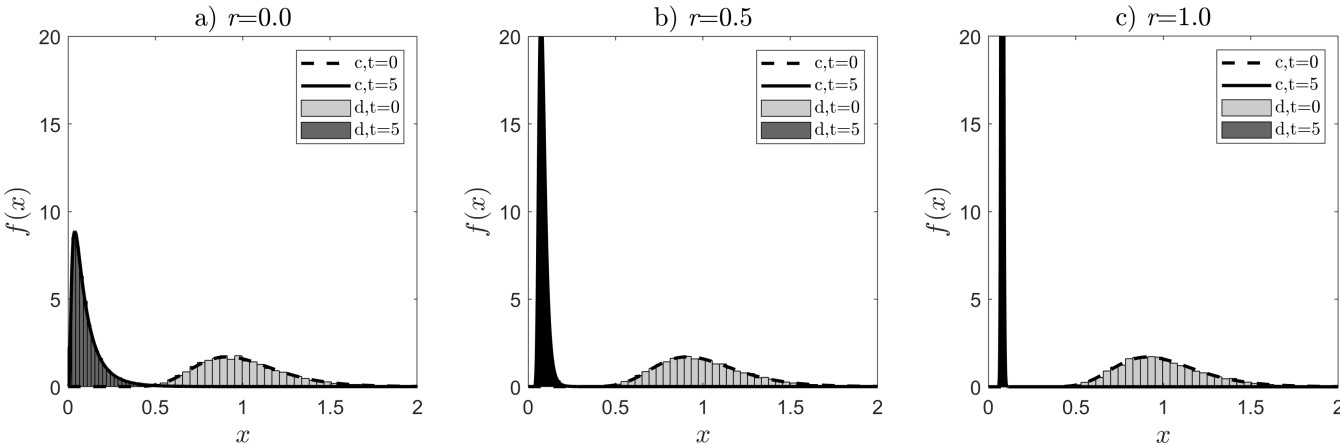

**Figure 4.** Evolution of a lognormal P.D.F in the compound kernel under the continuous (c) and discrete (d) models, at the parameter values $r = 0.0$ (left panel), $r = 0.5$ (middle panel) and $r = 1.0$ (right panel) from $t = 0$ until $t = 5.0$. The results of the discrete simulations are given by the histograms, the output of the continuous model are given by dashed (initial distribution) and solid lines (final distribution). Observe the fair agreement between the discrete and the continuous models.

## 3.2 Fitted lognormal distribution

Although the lognormal distribution is certainly *not* invariant under the compound kernel (i.e. an initially lognormal density function does not remain lognormal in the evolution), however, mass distributions in later timesteps highly resemble lognormal dsitributions. To test this visual observation we fitted lognormal distributions to the computed mass distributions in the discrete simulations. The evolution of the two parameters (respectively denoted to $\mu$ and $\sigma$) of the lognormal distribution are given in Figure 5 at values of the parameter $r$. The criticality of $r = 0.5$ is obvious in this setting, too: while the initially lognormal distribution is almost invariant under the evolution at $r = 1/2$, the evolution of the parameters $\mu$ and $\sigma$ take an opposite direction in the parameter space for $r = 0.0$ and $r = 1.0$, respectively. The $95\%$ confidence levels of the fit confirm the visual intuition: the evolved distributions are close to lognormal: in practical applications an approximation with a lognormal distribution produces an acceptable error.

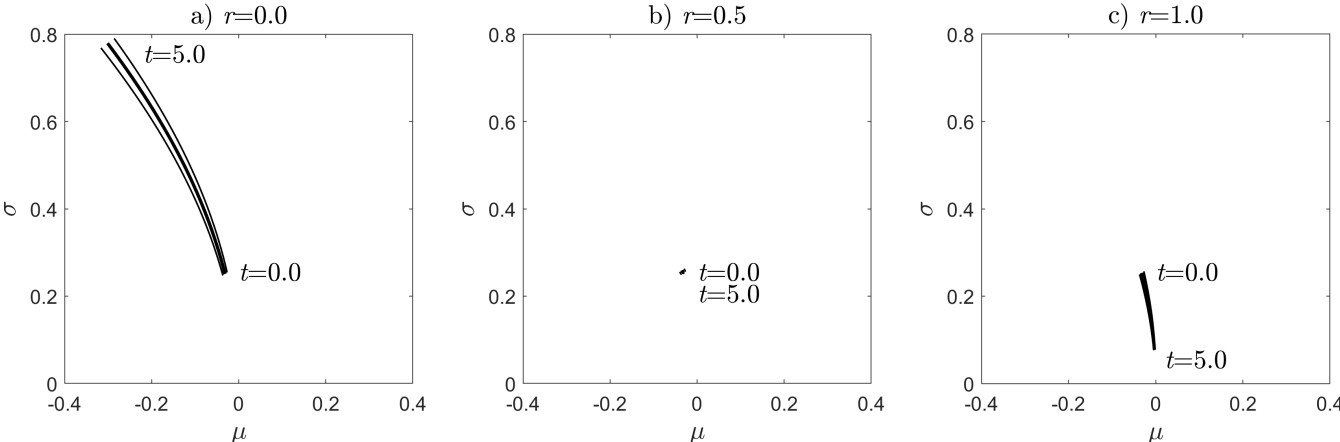

**Figure 5.** Parameters, $\mu$ and $\sigma$ of a lognormal distributions fitted to the computed mass distribution in the compound model, at the parameter values $r = 0.0$ (left panel), $r = 0.5$ (middle panel) and $r = 1.0$ (right panel) from $t = 0$ until $t = 5.0$. Thick solid lines correspond to the best fit, thin lines indicate the $95\%$ confidence level of the fit. Observe the narrow zone spanned by the confidence intervals.

### 3.3 Outliers: anomalies in smaller samples

The continuum model describes the $N \to \infty$ limit of the system. In the computations shown in Subsections 3.1 and 3.2 we either showed results based on the continuum model or, in the direct, discrete simulations we treated large ($N = 5000$) populations. However, if we look at the discrete simulations on smaller samples we may observe unexpected phenomena not recorded

5    in the previous computations. In Fig. 6 we show the mass distribution of a system at $r = 0.6$ with $N = 2000$ particles. The bulk of the histograms can be well approximated with a log-normal distribution. However, there are 12 particles with somewhat larger volume than predicted by the lognormal distribution and one approximately 150 times the median volume (5.3 times the radius). Thus inside the focusing regime we may observe a situation where we have a well-defined narrow distribution which describes the bulk of the particles but a few might escape from this process and may be left behind, at larger mass. This effect

10    is persistent and it was observed also for the parameter value of $r \simeq 0.7$.

In order to estimate the robustness of this scenario we use a simple approximation by assuming that all but one particles have volume $X$ and one single, exceptional particle, called 'outlier' has a volume $aX$ with $a \gg 1$. As it is demonstrated in Appendix E, the outlier can coexist with the population of the small particles. In the $N \to \infty$ limit the condition of such a coexistence reads

$$\frac{2a^r}{1+a} \leq 1. \tag{12}$$

Numerical solution of (12) for equality yields the *critical curve* $a_{\text{crit}}(r)$ on the $[r, a]$ parameter plane, separating systems where outliers may coexist with the population from systems where they may not. While we computed the $a_{\text{crit}}(r)$ critical curve for the case of infinitely large populations (the $N \to \infty$ limit), we stress the fact that the illustrated phenomenon is

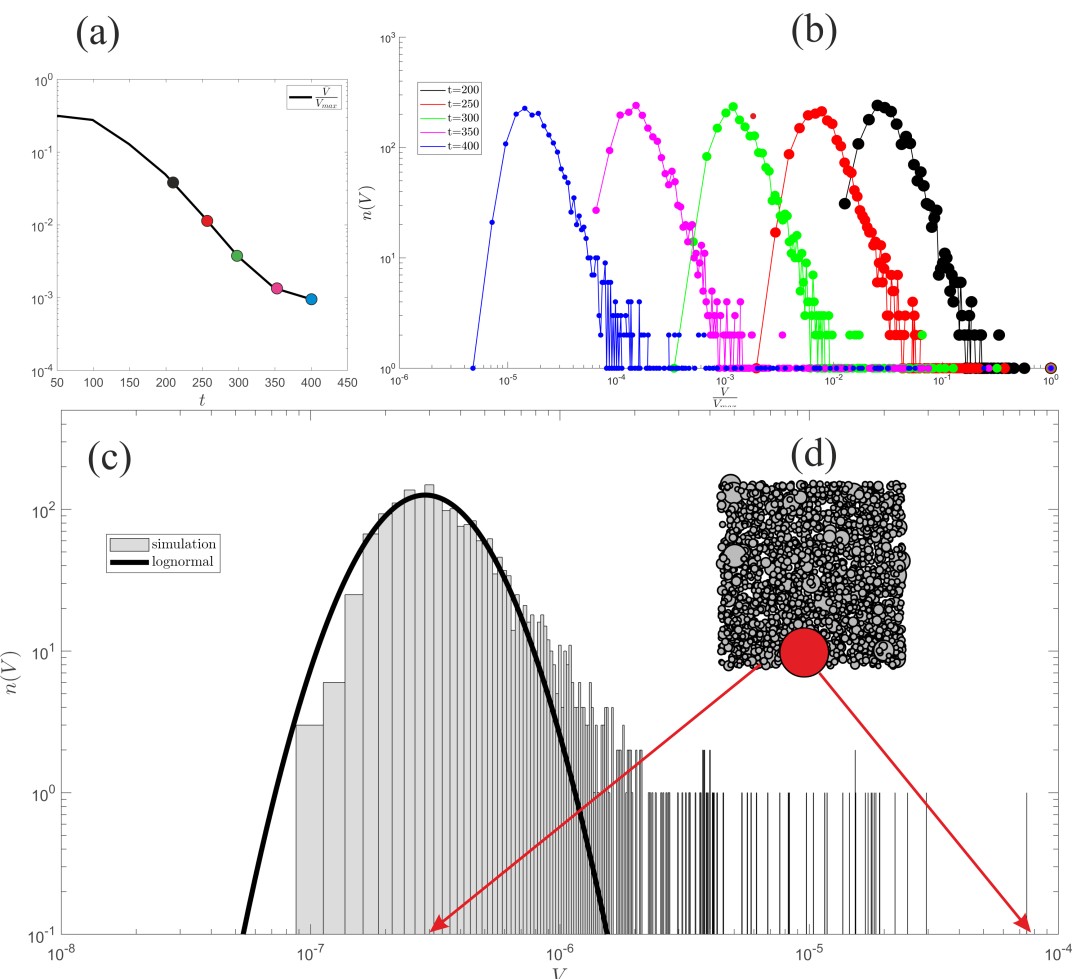

**Figure 6.** Simulation of a finite sample with $N = 2000$ particles. Inset (a) shows the evolution of the mean volume normalized by the maximal volume. Inset (b) depicts the evolution of the distribution, the corresponding points in (a) are denoted by the same color. The green curve ($t = 300$) is the one shown in detail in panel (c), it depicts the particle volume histogram after 300 collisions per particle. The grey boxes show the logarithmically binned histogram, the black line is a log-normal fit to the data. Observe the existence of outliers on the right. Inset (d) is a visual illustration of the entire population: all particles are placed randomly into a 2D container. Smaller particles were placed first and the white content (gray scale) is proportional to the linear size of the particle. One small particle close to the mean and one large particle (outlier) are marked with red and their position is indicated in the distribution.

inherently discrete and does not arise in the continuum model. We may explain this curious phenomenon in the following manner. Assume that we start from a narrow distribution. Then random fluctuations in the discrete system may create particles with large relative mass (i.e., a large parameter value $a$). If these fluctuations are sufficiently large to create particles *above* the critical curve $a_{\text{crit}}(r)$ then these outliers will be sustained, otherwise their mass will again approach the average mass of

the majority. The critical curve in Figure 7. shows that in the vicinity of the critical value $r = 0.5$ almost any such fluctuation will be sustained and outliers are likely to survive. However, as the parameter $r$ is increasing, it gets increasingly less likely to see sustained outliers. Another observation is that as the likelihood for the existence of outliers decreases, their expected relative size is increasing which matches the common-sense observation that the larger the outlier, the less frequently it may be observed. We also note that the relationship between the collection of small particles and the large particle is essentially asymmetrical. While the evolution of the latter is strongly influenced by both the factor $a$ and the control parameter $r$, the evolution of the density function for the small particles is solely controlled by the latter. In other words, adding one (or a few) very large particles to a collection of many small particles will not alter the fate of the latter, as long as the collisions between a pair of particles are based on a uniform choice.

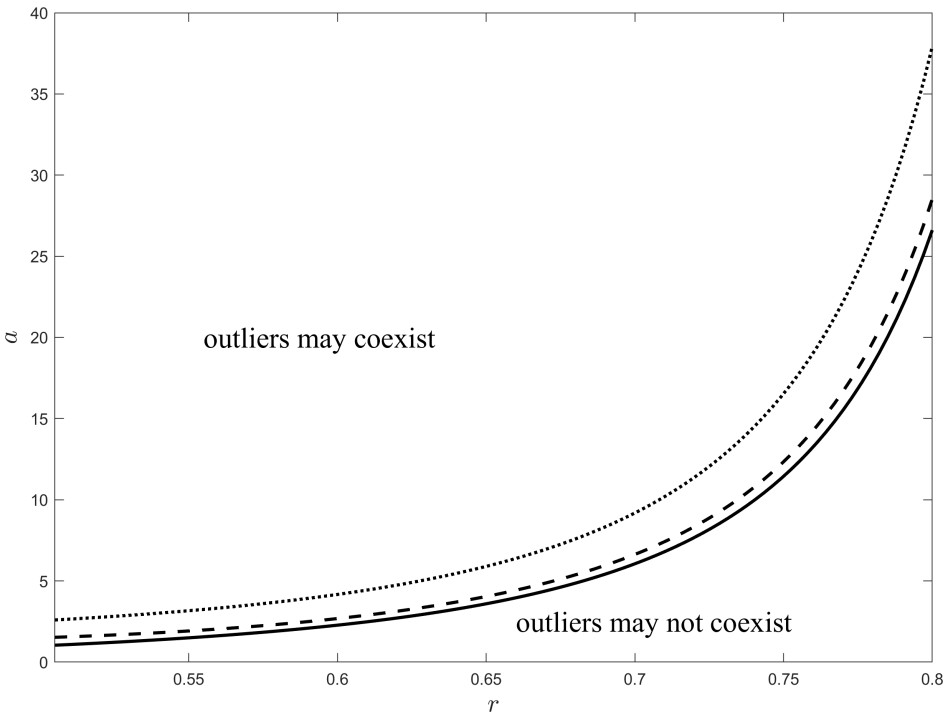

**Figure 7.** Critical curve $a_{\mathrm{crit}}(r)$ on the $[r, a]$ parameter plane. Systems with parameters $(r, a)$ associated with points above the curve admits the coexistence of outliers while the systems associated with points below the critical curve do not admit the coexistence of outliers. Solid line belongs to the $N \to \infty$ limit, dotted line represents $N = 20$ and the dashed line $N = 100$ particles, respectively.

## 4    Conclusions

In this paper we presented the first statistical model for the collective mass evolution of pebble populations under collisional abrasion. While our model is certainly not unique, it is compatible with

(a) existing geological observations,

(b) existing geometrical theory of individual and binary abrasion of pebbles,

(c) existing theory for individual mass evolution of pebbles (Sternberg's Law) and

(d) exiting statistical theory of coagulation and fragmentation.

In the spirit of standard statistical theory for collective evolution, our model is based on two components: (i) the binary collision kernel and, based on the latter, (ii) the governing equation for the evolution of probability density functions for mass distribution. Regarding (i) we used the model from (Domokos and Gibbons, 2018) which incorporates the existing theory for individual and binary abrasion, regarding (ii) we used the Fokker-Planck equation which is broadly used in the theory of coagulation in fragmentation.

Our collision kernel includes the single scalar parameter $r$ which can be associated with the energy level of the collective collisional evolution process. We found that $r = 0.5$ is *critical*, separating two regimes with fundamentally different behaviour: for $r > 0.5$ (low energy regime) we found *focusing* behaviour with *decreasing relative variance $R(t)$* and for $r < 0.5$ (high energy regime) we found *dispersing* behaviour with *increasing relative variance $R(t)$*. In geological terms, this result suggests that in low energy environments collisional abrasion acts on mass distributions in unison with size-selective transport while

in high energy environments the opposite happens and the two processes are counter-acting. In accordance with prevailing geological observations and Sternberg's Law,, our models predicts exponential decay of particle mass in both energy regimes.

    We investigated our model on two levels: (i) as a continuum model by regarding the evolution of the Fokker-Planck equation and (ii) as a discrete model by running discrete event-based simulations. In case of the continuum model we derived our results analytically and also from numerical simulation of the Fokker-Planck equation while in the discrete model we relied

on numerical computations. In regard of the existence of the critical parameter $r = 0.5$ and the existence of the focusing and dispersing regimes, the two approaches yielded quantitatively matching results.

    Large boulders among many small pebbles are often visible in mountain ranges of rivers. While this phenomenon is commonly attributed to transport, our model suggests that under some conditions, here again transport and abrasion may act in unison: we identified a curios phenomenon *not* present in the continuum model but present in the discrete model (even in the

25 $N \to \infty$ limit). If the parameter $r$ was in the focusing $r > 0.5$ range but not very far from the critical value $r = 0.5$, the bulk of the distribution was narrowing (in accordance with our analytical predictions), however, we could also observe a few particles with substantially larger mass (outliers), escaping the bulk of the distribution. We characterized the mass ratio of outliers versus the mean of the bulk distribution by the parameter $a$ and we derived a *critical curve $a_{\mathrm{crit}}(r)$* separating systems where outliers may be observed from those where this may not happen. Our result predicts that larger outliers are less likely to be observable.

While our paper only dealt with on size distributions, however, there exist also related observations on shape: sharp peaks in distributions of axis ratios (also referred to as *equilibrium shapes*) are mentioned in Bluck (1967); Dobkins and Folk (1970); Landon (1930); Orford (1975); Williams and Caldwell (1988); Ashcroft (1990); Lorang and Komar (1990); Yazawa (1990); Wald (1990). In Domokos and Gibbons (2012) a plausible argument was presented that equilibrium shapes may emerge on

shingle beaches as the result of interaction of abrasion and transport. We hope that the extension of the statistical theory presented in this paper may be capable to verify these observations.

**Appendix A: Testing the model for heterogeneous pebble populations**

In the context of the binary evolution model (1)-(2) we introduced the binary abrasion parameters $c_{12}$ and $c_{21}$ and for simplicity (since we only aimed to treat homogeneous populations) we used the same notation in the collision kernel (3)-(4). Here we refine this concept in the statistical setting for heterogeneous populations where we regard the collective evolution of $N$ particles with $M \leq N$ different materials $m_i$, $(i = 1, 2, \ldots M)$. (The binary case corresponds to $N = 2$, if the two pebbles are made from different material then we have $M = 2$ and for pebbles with identical materials we have $M = 1$. In the latter case in (1)-(2) we have $c_{12} = c_{21} = c$.)

In the statistical setting the binary abrasion parameters can be organized into an $M \times M$ matrix $\mathcal{M}$ with entries $c_{ij}$, $(i, j = 1, 2, \ldots M)$. The binary parameter $c_{ij}$ is defined as the constant coefficient in the collision kernel (3)-(4) associated with the abrasion rate of a particles with material $m_i$, bombarded by particles with material $m_j$. Needless to say, the matrix $\mathcal{M}$ is not symmetrical, in general we have $c_{ij} \neq c_{ji}$. In particular, if material $m_i$ is much harder than material $m_j$ then we expect $c_{ij} \ll c_{ji}$.

Based on the above considerations, the statistical model is controlled by the $M \times M = M^2$ binary abrasion parameters and the single environmental parameter $r$. Testing this model can be done along the strategies outlined in subsection 1.4 for homogeneous populations, however, more detail has to be observed.

(a) One may test the model at the *input level*, by fitting the kernel (3)-(4) to laboratory tests for pair-wise selected materials $m_i, m_j$. In such a test the abrasion rate of particles of material $m_i$ under abrasion from particles of material $m_j$ is plotted as a function of particle size of the abraded particle (with material $m_i$). Such experiments can be used to determine the binary abrasion parameters $c_{ij}$ for a given heterogeneous population. If the laboratory test is imitating the environment of the natural process, the environmental parameter $r$ may be also obtained in this manner. We will show such an example below.

(b) One may test the model at the *output level* by measuring the time evolution of full mass distributions and fitting the $M^2$ material parameters $c_{ij}$ and the environmental parameter $r$ to this dataset.

Next we show an example for testing the model at the input level by using the data obtained in Attal and Lavé (2009). Here the authors report on flume experiments where they measured the abrasion rate $E_d$ of individual limestone gravels with diameter between $D = 9$ and $D = 39$ mm mixed in approximately 400g of 10-18 mm and 18-28 mm granitic gravel. In our terminology, we have $M = 2$ (two materials) and we will use $m_1$ for the limestone and $m_2$ for the granite. The joint evolution of such a heterogeneous population is described by $M^2 = 4$ binary material constants: $c_{11}, c_{12}, c_{21}$ and $c_{22}$. The authors were primarily interested in the abrasion rates for limestone and they produced the $E_d(D)$ plots for these particles. In this experiment we may assume that the abrasion of the limestone pebbles was exclusively due to collisions with the granitic gravel (i.e. we

disregard limestone-limestone collisions). Thus the only relevant collisions are between limestone and granite, and for the mass loss $X(t)$ of the limestone we will use equation (3) with $X$ and $Y$ denoting the volumes of the colliding limestone and granitic particles respectively, and $c_{12}$ denoting the binary abrasion parameter associated with limestone being abraded by granite. (Not reported, by we may assume $c_{21} \ll c_{12}$). If we replace the volume of the granitic particles by their average the abrasion rate as

5  function of its diameter can be calculated numerically. Note that the abrasion rate $E_d$ in our notation reads

$$E_d = -\frac{X_t}{X}. \qquad (A1)$$

We fitted equation (3) to the dataset provided in Attal and Lavé (2009). We minimized the mean square error (with respect to the results in Attal and Lavé (2009)) for the parameters $r$ and $c_{12}$ and obtained $r = 0.19, c_{12} = 0.28$. Our fitted curves are illustrated in Figure A1 showing fair agreement between the data and the fitted model. The value of the environmental

10  parameter is in the range where we expect dispersing behaviour, as we discussed in Appendix D, which is in accordance with the target of the original experiment which simulated abrasion in fluvial environments. We note that the same parameter-pair $r = 0.19, c_{12} = 0.28$ is valid for both limestone experiments (i.e. these parameters do not depend on the size of the particle). Our fit appears to be consistent in this respect.

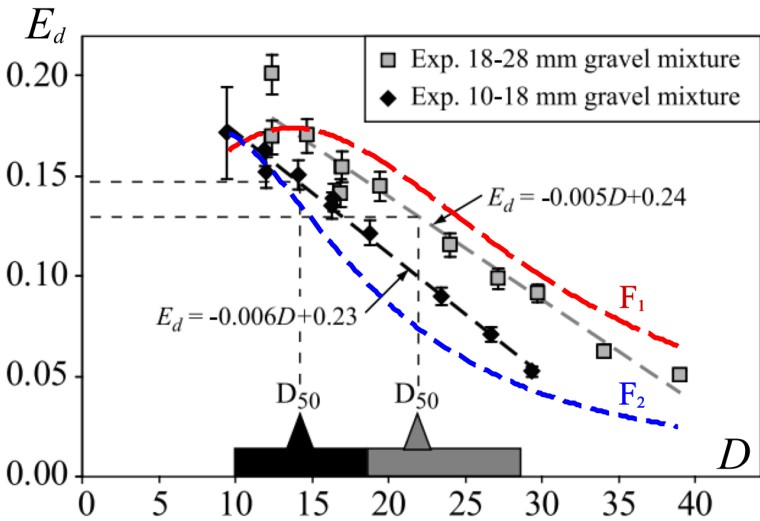

**Figure A1.** (a) Abrasion rate $E_d$ predicted by the compound kernel (3) fitted to experimental data in Attal and Lavé (2009). Figure 9a by Attal and Lavé (2009) superposed with our model fits $F_1$ and $F_2$. Mass estimated from diameter. Least squares optimization yields $r = 0.19$ and $c_{12} = 0.28$.

## Appendix B: Some properties of the kernels in Subsection 2.3

### B1 Summation kernel

Differential equations governing the time evolution of the first and second moments can be readily obtained, hence the mean $E^+(t)$ and variance $W^+(t)$ follows the following initial value problems (IVPs):

$$E_t^+(t) = -2E^+(t) \quad \text{with} \quad E^+(0) = E_0, \tag{B1}$$

$$W_t^+(t) = -2W^+(t) \quad \text{with} \quad W^+(0) = W_0. \tag{B2}$$

It follows, that both the expectation and the variance exhibit exponential decay, namely $E^+(t) = E_0 e^{-2t}$ and $W^+(t) = W_0 e^{-2t}$. It is straightforward to show, that the relative variance $R^+(t)$ increases exponentially

$$R^+(t) := \frac{W^+(t)}{E^+(t)^2} = \frac{W_0}{E_0^2} e^{2t} = R_0 e^{2t}. \tag{B3}$$

### B2 Product kernel

In case of the product kernel the IVPs describing the evolution of the mean $E^*(t)$ and variance $W^*(t)$ respectively read

$$E_t^*(t) = -(E^*(t))^2 \quad \text{with} \quad E^*(0) = E_0, \tag{B4}$$

$$W_t^*(t) = -2W^*(t)E^*(t) \quad \text{with} \quad W^*(0) = W_0. \tag{B5}$$

Here the decay of the mean and the variance are polynomial as we find

$$E^*(t) = \frac{1}{t + \dfrac{1}{E_0}}, \quad \text{and} \quad W^*(t) = \frac{W_0}{E_0}^2 \frac{1}{\left(t + \dfrac{1}{E_0}\right)^2}. \tag{B6}$$

which result in a steady relative variance $R^*(t)$, determined by the initial distribution $f_0$. In specific

$$R^*(t) := \frac{W^*(t)}{E^*(t)^2} = \frac{W_0}{E_0^2} = R_0. \tag{B7}$$

## Appendix C: Approximate investigation of the compound kernel

### C1 Truncated compound kernel

The truncated compound kernel is obtained from the compound kernel as the truncated Taylor polynomial computed at $y = x$ with an $(O(y - x)^2)$:

$$\psi^{c,T}(x,y) := \frac{x}{2} + \left(\frac{1}{4} - \frac{r}{2}\right)(y - x) + O((y - x)^2). \tag{C1}$$

Using the master equation, the following Cauchy problems are found that define the evolution of the mean and the variance:

$$E_t^{c,T}(t) = -\frac{1}{2} E^{c,T}(t) \qquad \text{with} \qquad E^{c,T}(0) = E_0, \tag{C2}$$

$$W_t^{c,T}(t) = -\left(\frac{1}{2} + r\right) W^{c,T}(t) \quad \text{with} \quad W^{c,T}(0) = W_0. \tag{C3}$$

Solution of these ODEs yields the evolution of the relative variance as

$$R^{c,T}(t) := \frac{W^{c,T}(t)}{E^{c,T}(t)^2} = \frac{W_0}{E_0^2} e^{\left(\frac{1}{2} - r\right)t}. \tag{C4}$$

## C2 A population of identical particles preserved

Here we show, that a population of identical particles, characterized by a Dirac delta function as input P.D.F. is preserved in the model with the compound kernel regardless of the value of parameter $r$. Without loss of generality, we investigate the evolution from the $f_0 = \delta(1)$ initial condition, where $\delta(x)$ denotes the Dirac-delta function at $x$. Obviously, $E_0 = 1$ and $W_0 = 0$. We show, that now $f(t,x) = \delta(c(t))$ holds for any $t > 0$. Let us assume, that at some $t^* \geq 0$ the distribution is $f(t^*,x) = \delta(c(t^*))$ with mean $E^c(t^*) = c(t^*)$ and variance $W^c(t^*) = 0$, respectively. Observe that

$$\int_0^\infty f(t^*,y)\psi^c(x,y)dy = \psi^c(x,c(t^*)). \tag{C5}$$

The time derivative of the mean can be computed via

$$E_t^c(t^*) = \int_0^\infty f_t(t^*,x)x\,dx = -\int_0^\infty f(t^*,x)\psi^c(x,c(t^*))dx = -\frac{1}{2}c(t^*), \tag{C6}$$

where we used (7), applied integration by parts and employed (C5). Similarly, the evolution of the variance is found to follow:

$$W_t^c(t^*) = \int_0^\infty f_t(t^*,x)x^2\,dx - 2E_t^c(t^*)E^c(t^*) = -2\int_0^\infty f(t^*,x)\psi^c(x,c(t^*))x\,dx + c(t^*)^2 = -c(t^*)^2 + c(t^*)^2 = 0. \tag{C7}$$

This shows that the variance of the distribution is constant, and as it started at $W_0 = 0$, i.e. it vanishes whole along the evolution. In other words, the we have a Dirac-delta (degenerate) distribution at any $t \geq 0$. Employing (C6) we find, that the location $c(t)$ follows the initial value problem $c_t(t) = -\frac{1}{2}c(t)$ with $c(0) = 1$, hence $c(t) = \exp(-\frac{t}{2})$.

## C3 Dispersing and focusing behaviour identified in the population of almost identical particles

As the model lacks diffusion, the behaviour of a degenerate distribution with all the mass concentrated at a single value is worthy to study, because long term existence of a set consisting of identical particles can take place in the model. In Appendix C2 we show that a population of identical particles remain identical in our model. In other words, time-invariance of the Dirac-delta distribution holds in our model, regardless of the value of the parameter $r$. Nevertheless, the value of $r$ affects the *stability* of that Dirac-delta: next we show that the evolution for a population of *almost* identical particles (i.e. a perturbed version of

the Dirac delta distribution) is either focusing or dispersing, depending on the value of $r$. To see this, we define a perturbed distribution. Let $\varepsilon > 0$ be a fixed parameter and define

$$\hat{f}_0(x) := \begin{cases} (1-\varepsilon)\delta(1) + \frac{1}{2} & \text{if } 1-\varepsilon \le x \le 1+\varepsilon \\ 0 & \text{otherwise .} \end{cases} \tag{C8}$$

It is straightforward to show that $\int_0^\infty \hat{f}_0(y)\psi^c(x,y)dy = \psi^c(x,1)$, $E_0^c = 1$ and $M_2^c(t) := \int_0^\infty f(t,x)x^2 dx$ with $M_2^c(0) = 1 + \frac{1}{3}\varepsilon^3$. We aim to investigate the sign of $R_t^c$ at $t = 0$. Since $R^c(t) = M_2^c(t)E^c(t)^{-2} - 1$, we need to study the sign of $M_{2,t}^c(0)E^c(0) - 2E_t^c(0)M_2^c(0)$. Integration by parts yields

$$E^c(0)M_{2,t}^c(0) - 2M_2^c(0)E_t^c(0) = -2\int_0^\infty \hat{f}_0(x)\psi^c(x,1)x\,dx + 2(1 + \frac{1}{3}\varepsilon^3)\int_0^\infty \hat{f}_0(x)\psi^c(x,1)dx = \frac{1}{3}\varepsilon^3\left(\frac{1}{2} - r\right) + O(\varepsilon^3), \tag{C9}$$

where algebraic manipulations leads the last equality. In accordance with the results on the truncated model, we found, that $r = 1/2$ is critical, at $r < 1/2$ the relative variance $R_t^c$ is positive, it increases for any $\varepsilon > 0$, i.e. the population of identical particles is *unstable*, small perturbations disperse the mass distribution. At $r > 1/2$ the relative variance $R_t^c < 0$, which shows, that the population of identical particles is *stable*, the model is focusing.

## Appendix D: Estimating physically possible values of $r$

In the paper we assumed that the particle collision probability depends on the volume of the particles as

$$P(X) \propto X^r. \tag{D1}$$

Here we investigate two extreme scenarios, associated with the collision probabilities $P_{\text{smooth}}(X)$ and $P_{\text{turbulent}}(X)$ where we expect $r$ to assume its extreme values.

The first is the smooth gradient flow. In such a case the driving fluid has a strong, but on a particle-size scale constant velocity gradient in one of the spatial directions. Such situations may arise, e.g., in shallow water layers. Here the relative velocity of the particles grows with the distance. If we are at distance $u$ from the center of the particle in the direction of the flow velocity gradient, the collision probability $P_{\text{smooth}}(X)$ can be estimated by the product of the velocity difference and the linear cross-section of the particles. (Note that $R \equiv X^{1/3}$ is the linear size of the particle):

$$P_{\text{smooth}}(X) \sim \frac{1}{R}\int_0^R u\sqrt{R^2 - u^2}\,du = \frac{1}{3}R^2 = \frac{1}{3}X^{2/3}. \tag{D2}$$

Based on (D1), this gives us an estimate for high $r = 2/3$.

The other extreme case is a fully chaotic motion where equipartition takes place (Uberoi, 1957). Thus the kinetic energy of the particles ($\frac{1}{2}\rho X v^2$) is independent of their volume. Thus the speed of the particles must be proportional to $X^{-1/2}$. If

we disregard correlations the particles have a cross-section proportional to their projected area which is proportional to $X^{2/3}$. Combining the two gives us

$$P_{\text{turbulent}}(X) \sim X^{-1/2} X^{2/3} = X^{1/6} \tag{D3}$$

and based on (D1) we obtain $r = 1/6$. Thus it is possible to have physical scenarios apparent in nature where the value of $r$ falls to either side of the critical value of $r_c = 0.5$ with large enough margin.

## Appendix E: Investigation of outliers in finite samples

Let us have a sample with $N$ particles with $(N-1)$ having identical volume $X$. The last particle is an outlier with volume $aX$, where $a \ll 1$. In a single binary collision, a hit between particles with volume $X$ is called an A-type event, while a collision with the outlier being involved is a B-type event. Based on discrete probabilistic considerations, the probability of an A-type event equals $(N-2)/N$ and a B-type event is $2/N$, respectively. In the A-type event the average size $\bar{X}$ of the particles with volume $X$ after the collision that lasts for $\Delta t$ reads

$$\bar{X} = \frac{2(X - X/2\Delta t) + (N-3)X}{N-1} = X - \frac{X}{N-1}\Delta t. \tag{E1}$$

Computing $aX/\bar{X}$ and truncating the Taylor series expansion in $\Delta t$ after linear terms around the value $\Delta t = 0$ yields the time derivative of the parameter $a$ associated with an A-type event:

$$a_t^A = \frac{a}{N-1}. \tag{E2}$$

In case of the B-type event both the outlier and one of the small particles follow the compound kernel via:

$$(aX)_t = -\frac{a^{1+r}X}{1+a}, \tag{E3}$$

$$X_t = -\frac{a^{1-r}X}{1+a}. \tag{E4}$$

The second equation is employed to compute the average volume of the small particles (i.e., $\bar{X}$ associated with this event). Now we need to truncate the Taylor series of $\frac{aX - (aX)_t \Delta t}{X - \bar{X}}$ at $\Delta t = 0$. After algebraic manipulations we find

$$a_t^B = -\frac{a^{1+r}}{1+a} + \frac{a^{2-r}}{(1+a)(N-1)}. \tag{E5}$$

Considering the probabilities of events A and B we arrive to

$$a_t = a\frac{N-2}{N-1} - 2\left(\frac{a^{1+r}}{1+a} + \frac{a^{2-r}}{(1+a)(N-1)}\right). \tag{E6}$$

Note that an increase in the value of $a$, i.e., $a_t > 0$ implies that the outlier is getting further from the population. In the case of the $N \to \infty$ limit we find

$$a_t = a\left(1 - \frac{2a^r}{1+a}\right). \tag{E7}$$

Here the sign of the expression in the brackets determines the sign of $a_t$, which coincides with e.q. (12) in the text. One can also show, that if there exist $a_\text{crit} > 1$ such that $a_t = 0$ at $a_\text{crit}$, then $a_t > 0$ for any $a > a_\text{crit}$. Hence, we need the $a_\text{crit} > 1$ that makes the expression in the bracket vanish. Existence of such a critical value can be shown for the case with finitely many particles, too. As we can see, sufficiently large outliers may with the population on the long run. The control parameter $r$ determines

5    how large an outlier needs to be for sustained coexistence.

*Author contributions.*  G.D. proposed the problem and supervised the research; A.Á.S. carried out the analytical and numerical study of the continuous model, T.J. developed the discrete numerical model; G.D., A.Á.S. and J.T. wrote the paper.

*Competing interests.*  The authors declare that they have no competing interests.

*Acknowledgements.*  The authors are indebted to the referees D. Furbish, S. Carretier and D. Bertoni. Responding to their commenst helped

10    to improve the manuscript substantially. The authors sincerely thank J. Lavé and M. Attal for sharing the original dataset of their abrasion experiments. The research reported in this paper was supported by the BME Water Sciences & Disaster Prevention TKP2020 IE grant of NKFIH Hungary (BME IE-VIZ TKP2020) and the NKFIH grant K134199.

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
