# Peer review of "Particle size dynamics in abrading pebble populations"

_Earth Surface Dynamics, 2020_

## Short Comment (SC1) · 2 Nov 2020

Review of:

**Particle size dynamics in abrading pebble populations**

András A. Sipos, Gábor Domokos and János Török

**General Comments**

Very clever. Lovely.

This paper merits publication in *Earth Surface Dynamics* without too much ado. I suggest that revisions be focused on providing points of clarification for the *ESD* readership.

The centerpiece of the paper is its use of a Master equation, in the form of a Fokker-Planck equation, to describe the evolution of the probability distribution of particle states (sizes) in relation to abrasion due to particle collisions. A particularly interesting feature of the formulation is that the Fokker-Planck equation only involves advection. This occurs because the embedded kernel explicitly treats all possible changes in state associated with binary collisions involving each particle size with all other sizes, such that "diffusion" is unnecessary (but could be added depending on the physics involved). The formulation thus naturally leads to the ideas of *dispersion* versus *focusing* in describing the evolution of the form of the distribution of particle sizes, notably its variance — which represents a particularly interesting (and I think novel) outcome of the analysis. It is refreshing to see work that highlights how variability about the expected value is as important as the expected value in describing the dynamical behavior of the system.

Here I am compelled to offer a philosophical point regarding the value of the formalism provided by the Master equation and the Fokker-Planck equation. If we accept the veracity of the classical interpretation of probability (Hájek, 2012), then the Master equation is an element of physics (statistical mechanics) that is as close as it gets to the idea of "truth" that we normally reserve for the cumulative knowledge of mathematics. That is, it is a precise probabilistic ledger of admissible particle states and changes in these states. Then, the kernel within the Master equation (and the Fokker-Planck equation) contains the physics that govern the changes in state — just as, for example, the joint probability distribution of particle travel times and travel distances within the entrainment form of the Exner equation (a specialized Master equation) contains the essential physics of particle motions. I therefore view this paper as a particularly important contribution to the conversation on sediment particle comminution. The Master equation (Fokker-Planck equation) provides a wonderful framework. The path forward then involves further unfolding the physics of the kernel involved. The authors focus on a particle-mass-based kernel that is faithful to semi-empirical results regarding particle comminution (e.g., the exponential decline in the expected particle size). Perhaps eventually the physics can be elaborated to more explicitly incorporate, for example, particle collision energetics during downstream transport.

**Specific Items**

**Figure 1 and accompanying text:** The problem is conceptualized in terms of abrasion due entirely to particle-particle collisions. Because of this I anticipate the possibility of unease among some readers.

Specifically, each of the kernels, Eq. (5), Eq. (6) and Eq. (7), involves the masses $X$ and $Y$ of the *abraded* particle and the *abrader* particle. (The couplet in Eq. (1) clarifies that, with binary collisions, both particles are abraded and abrader particles.) This formulation thus essentially envisions binary collisions amongst a cohort of (continuously moving?) particles with a defined size distribution, and it excludes effects of particle-bed collisions. Inasmuch as a sediment bed is composed of particles and these are nominally included in the distribution of sizes, then strictly speaking all collisions involve particles with well-defined masses. But in natural and experimental settings, abrasion also involves particle-bedrock collisions. Moreover, inasmuch as bed particles are relatively immobile, then they act the same as bedrock with respect to an impacting particle. In this situation, the mass of the impacted "particle" (abraded? abrader?) is effectively infinite, such that the collision kernels are not meaningful. This issue becomes increasingly important as transport becomes rarefied, where effects of particle-bed collisions dominate over particle-particle collisions — akin to granular shear flows at high Knudsen number (Kumaran, 2005, 2006). Rarefied conditions represent an interesting limiting case. One might presume that the kernel involves only the mass of the impacting particle such that the Fokker-Planck equation takes on its ordinary form, but with inhomogeneous drift whose effect is focusing (depending on the value of $r$)?

That said, I raise these points to suggest the need for further clarity rather than as a criticism of the formulation. I view this work as an important, well-crafted brick designed to fit into the edifice of our understanding of particle comminution — *sensu* Forscher (1963). The selected kernels are a starting point. Effects of particle-bed impacts can wait for later.

Second, the authors comment on the idea that binary collisions constitute the basis of the collision kernel such that with large $N$, collective size evolution is driven by many binary collisions. I would add that, in fact, binary collisions are physically the correct choice. The formulation is continuous in time, and for this situation the probability that an individual particle experiences collisions with two or more particles simultaneously during an infinitesimal interval $dt$ is vanishingly small if not zero. (Binary collisions of course are a foundational element of kinetic theory.)

Here is an idealized visual for what the formulation is describing. Envision a large container containing a great number $N$ of particles (sort of like Figure 6d). We continuously, vigorously shake the container giving particle-particle collisions together with particle-container-wall collisions. For large $N$, effects of particle-particle collisions dominate over particle-container-wall collisions to produce abrasion. (This is directly reflected in the ODE form of Eq. (1), such that the formulation does not address downstream motion — which is entirely fine for starting the conversation.) Indeed, this visual is reminiscent of experimental abrasion studies that have in fact used containers of particles ("tumbling" experiments) to describe particle comminution. And, the shaking of a container with a great number of particles is essentially the basis of numerical simulations of granular gases (which have

revolutionized this field).

**Section 2.2:** The authors are fully aware of the points made in the following comments, so these are merely offered in aiming at clarification of the presentation.

The quantities $X$ and $Y$ (and $x$ and $y$) physically represent the same thing. In writing $f_t(t,x)$ on the left side of the Master equation, Eq. (4), this denotes that $X$ is the size of interest — which experiences collisions with all other sizes $Y$ via the collision kernel. That is, if $X$ is the size of interest, then $Y$ is a simple way of denoting all other sizes. It is then noted that, "contrary to the majority of Fokker-Planck models, spontaneous fragmentation is not allowed in our model, eliminating diffusion." First, the effect of fragmentation in this type of formulation would not actually be represented by diffusion. As previously alluded to, the formulation is number conserving, and fragmentation thus is not admissible. Letting $h = 1/N$ denote the small amount of probability "carried" by each of the great number $N$ of particles, fragmentation of a particle of size $X$ means that this amount of probability is instantaneously "lost" from the interval $x$ to $x + \mathrm{d}x$ and partitioned into new amounts of probability $h = 1/(N{-}1{+}n)$ (where $n$ is the number of new fragments) which instantaneously "appear" in new intervals $\mathrm{d}x$ associated with smaller values of $x$. That is, fragmentation is described by local sink and source terms. (If other formulations indeed treat this as a diffusive process, then I would be skeptical of them.) Second, the actual reason that diffusion is not involved is because the size-loss kernel is deterministic (rather than treated as a random variable) *and* because the integral within the brackets in Eq. (4) is over all $y$. This is the same as saying that the number of particles $Nf(t,x)\mathrm{d}x$ within the small interval $x$ to $x + \mathrm{d}x$ during a small interval $\mathrm{d}t$ experiences binary collisions with all other possible sizes in proportion to their presence in the distribution. Because the kernel $\psi(x,y)$ is deterministic in $x$ and $y$, all possible changes in size of particles with starting size $x$ are explicitly accounted for so that a description of the rate of change in the variance of these changes in size — that is "diffusion" — is unnecessary. Hence the use of the words "dispersion" and "focusing" is accurate. If the kernel $\psi$ instead is treated as a random variable, then diffusion becomes involved.

As an aside: Starting with the more basic form of the Master equation, I am wondering what the Kramers-Moyal expansion, or its counterpart, looks like in getting to Eq. (4). (There is no need to show this; I'm inspired to figure it out.)

**Section 2.4, Lines 20–35:** This presentation in relation to flow conditions is, for me, a bit cryptic. For example, the authors appeal to a linear velocity profile for laminar flows, so I immediately imagine the case of Couette flow and wonder about its relevance. I also am not sure what "cross-section" is being referred to. I would recommend further explanation. It seems that the key point is that the different flow conditions suggest values of $r$ far from $r_{\mathrm{crit}} = 1/2$, leading to either dispersion or focusing behavior.

**Section 3.3:** This section is quite interesting. Based on a brief email exchange with Professor Domokos, I am suspecting that the authors will further elaborate the idea that the large particles ("outliers") in the numerical simulations involving finite $N$ (in contrast to the

continuum formulation) in a sense represent effects mentioned above — that particle abrasion in natural and experimental settings involves particle-bed/bedrock collisions, where the impacted "particle" (i.e., bedrock) has arbitrarily large mass. I also am envisioning situations in gravel bed streams where particularly large particles are present, consistent with the description in the Discussion.

**Appendix A4:** Once a Dirac function... always a Dirac function (for the compound kernel and regardless of the value of $r$). Of course! I love it!

I hope that my comments are useful to the authors.

djf

**References**

[1] Forscher, B. K.: Chaos in the Brickyard, *Science* **142**, 339 (1963).

[2] Hájek, A.: Interpretations of Probability, *The Stanford Encyclopedia of Philosophy* (Winter 2012 Edition), Edward N. Zalta (Ed.), http:// plato.stanford.edu/ archives/ win2012/ entries/ probability-interpret/ (2012).

[3] Kumaran, V.: Kinematic model for sheared granular flows in the high Knudsen number limit, *Physical Review Letters* **95**, 108001 (2005).

[4] Kumaran, V: Granular flow of rough particles in the high-Knudsen number limit, *Journal of Fluid Mechanics* **561**, 43–72 (2006).

---

## Referee Comment (RC1) · Duccio Bertoni (Referee) · 3 Dec 2020

General comments: The paper is well written and structured. The objectives are clearly outlined at the beginning and well assessed at the end of the paper. As a geologist, I find the mathematical approach to geology-related issues useful to get deeper insights on such matters, which I obviously always addressed from a process-driven point of view. However, I'm not qualified to evaluate the maths behind the methodologies of the present manuscript. I find the results consistent with my experience about sediment transport, abrasion and movement, I cannot rise any geological inconsistencies.

Specific comments: I have just a few observations/requests for the Authors: 1. for the most part the theory about sediment transport, movement activation thresholds, depo-

[Figure]

**ESurfD**

Interactive
comment

sition, etc. has often been linked to the studies of Hjulström and Shields, who provided two popular diagrams that constitute the basis for such analyses. I see that the Authors did not cite them in the Introduction, and I wonder why. Are they obsolete? Or not really useful for your scope? Anyways, I'd like to ask the Authors to comment on this aspect. 2. being a field-bound geologist, I must admit (embarrassed) that I'm forced to accept mathematical models as a sort of leap of faith. Therefore, I cannot separate the model results from what I do observe on the field. My request is simple, then: you state in the Conclusions that your results are compatible with existing geological observations. I kind of reverse this statement. What would you do to further back up your conclusions? Could you think of some way to confirm your results? Just giving some perspectives about this aspect would strengthen the paper from my point of view. 3. paragraph "1.1 Geological observations": I've always been inclined to think that transport effects on sediments would prevail in a unidirectional flow, while abrasion effects would increase when the processes leading to sediment movement are more chaotic, multi-directional, and less predictable. Could you please comment on that?

Technical corrections: 1. even though largely used, "shingle beach" is not a fully technical term. I would prefer the more general "coarse-clastic beach" to refer to a beach constituted by coarse sediments. 2. typo, Figure 2 caption: "... relative size variation of the a mass...". 3. typo, paragraph 2.3 Collision kernels, line 15: "... a trade-off between between physical..." 4. typo, paragraph 3.2 Fitted lognormal distribution, line 4: "... in the discrete dsitributions."

Hope this helps!

---

## Referee Comment (RC2) · Sebastien Carretier (Referee) · 10 Dec 2020

This is a stimulating article, which makes an interesting link between a statistical description of pebble collisions, the physics of their attrition and the evolution of the size distribution of river or beach pebbles. I found particularly promising the distinction between reducing or increasing the variability of grain sizes related to attrition as a function of a parameter, r, which can be related to river energy. However, in order for this paper to be read and used by those, like me, who do not eat the Fokker-Planck equation for breakfast every morning, I recommend substantial additions described below. I have no substantive criticism of the results presented. My suggestions are intended to broaden the scope of this article for non-physicists.

[Figure]

At the first order, the authors could increase the impact of their findings by better relying on and linking it more closely to geological and experimental observations. The authors claim that their model "fits" the geological observations, but I see no clear evidence of such a fit in their paper. Several assertions lack references (see specific comments), in particular concerning the relative role of selective transport and attrition in the evolution of grain size along a river. In fact, selective transport has been mainly advanced to explain or model downstream fining in rivers (e.g. Paola et al., 1992; Ferguson et al., 1996; Fedele and Paola, 2007; Whittaker et al., 2011 - see list of papers at the end of this review). In parallel, several authors have shown evidence of attrition to explain the variation in grain size (e.g. Brewer and Lewin, 1993; Attal and Lave, 2006; Dingle et al., 2017). These studies are worth mentioning and discussing to show that the results presented in this manuscript are consistent with geological observations.

Theoretical results could also be confronted with experimental data, which are rare and valuable. In addition to Kuenen (1956), Attal and Lavé (2009) presented pebble attrition experiments on a 1:1 scale, representing Himalayan rivers conditions. These experiments show in particular that the mass loss by attrition increases with particle velocity but is weakly dependent on particle size. As the parameter r can represent the degree of dependence between the attrition rate and the particle size, a discussion on the possible value of r in the experiments of Attal and Lavé (2009) would allow to link the theory, these experiments and the prediction of the evolution of the relative variability (focusing or dispersing) in a natural case like that of Himalayan rivers.

Statistical physics is used here in a very clever way to describe the phenomenon of attrition. However, statistical physics (and the Fokker-Planck equation) has also been used to describe the transport of grains in a river. In this case, advection-diffusion emerges from a combination of probability densities that describe the distribution of the transport distances of the particles and that of their residence time at rest (e.g. Lajeunesse et al., 2018; Nikora et al., 2002). Longer residence times for larger particles (selective sorting) may explain some of the anomalous scattering observed in

tracer data (Phillips and Jerolmack, 2014; Carretier et al., 2019). In a natural system that combines both the statistical physics of transport and abrasion, how could the proposed theory be verified on the basis of field observations? What experiments or observations need to be carried out to distinguish between these two components? Again, linking this proposed theory with predictions that could be made by geologists or experimenters would increase the scope of this paper. I recommend significantly increasing the discussion in this direction.

In the discussion, always with the aim of linking this theory with the field, I suggest discussing the following points: The pebbles impact each other but also the bedrock (which could represent a much larger volume for the population y). How would this change the result? The lateral contribution of the hillslopes modifies the initial distribution fo(x) in the real case (e.g. Attal and Lavé, 2006; Sklar et al., 2017). Can we find a natural case where the proposed theory could be tested (canyon with one particular lithology at some localized point upstream, providing tracer pebbles that could be followed downstream for example)?

The structure of this manuscript is confusing: the conclusions are described in several details in the introduction. This has the merit of setting reader's minds, but when reading it for the first time, I wondered whether it was a reminder of previous work or new findings. To guide the reader, I think it is important to write clearly what the addressed question is in the introduction.

Specific comments

Page 2 Geological observations: extend this part based on above comments.

Page 2 Line 27-29: gives references.

Page 2 "as we can see": not obvious to me.

ÂăP3 L14: " in (Domokos and Gibbons, 2013) " wrong citation format for this reference throughout the text: in Domokos and Gibbons (2013)

P4 Equation for Xt, Yt: number the equations. Give the numbers of these equations in the original paper and the corresponding notations, to help the reader.

P5 L9 Useful assumption to avoid dealing with the diffusion term in the F-P equation, but the experiments of Attal and Lave (2009) show that fragmentation can be significant.

P5 L21 "setting this is not the case": Why?

P5 L31 "that the that the"

P6 L10: Is the correspondence with Sternberg's law demonstrated in this paper or in a previous paper?

P7 L6-10. I think it would be useful to discuss the meaning of the two populations. Are they particles of the bedload? Of the suspended load? Both? Does the collision model reflect a dynamic where the grains are in suspension? In saltation? I imagine that the physics of collisions is different in these cases.

P8 L7. Could you give a "hands-on" explanation of the nature of diffusion related to fragmentation in this case?

P8 L25 Summary or summation ?

P8 L26-30 Explain why you are testing these other kernels whereas you announce in the introduction that the kernel finally used is "the right one".

P L12: Proportional TO projected ? (could you give a reference?)

P9 L21-27 Give references or explain better the values deduced for r. Again, do these r values depend on the type of transport (suspension, saltation etc.)?

P9 L 26-27 Important to make the link with the field. Expand on this aspect.

P9 L32 what is V? (Volume I think but I have not seen its definition above).

P11 L10 justify why you take a log-normal law for fo. Does the result change if you take

another law?

P11 L11: "The applied time step is fixed at $\Delta t = 0.01$." Already said above.

P11 Figure 4: Indicate c: continuous and d: discrete in the caption. Because the by-product of abrasion is removed, I would have expected that the distribution tends toward a Dirac at x=0 in all the cases. Why is it not the case? ("hands-on" explanation please;). In the focusing case, what determines the final x value of the Dirac?

P12-12 3.3 Very interesting discussion. In Figure 4 it could also be argued that the tail of the distribution between 2E-7 and 2E-6 is a power law, while the outliers represent a subsampling of the actual distribution, as often observed in seismology (for magnitude) or hydrology (for peak discharge). Could the persistence of these outliers indicate some fractional derivative component not taken into account in the Fokker-Planck equation?

P14 "(a) existing geological observations": not demonstrated in my opinion.

Good luck with the revision,

Sebastien Carretier

Attal, M., and J. Lavé (2006), Changes of bedload characteristics along the Marsyandi River (central Nepal): Implications for understanding hillslope sediment supply, sediment load evolution along fluvial networks, and denudation in active orogenic belts, Spec. Pap. Geol. Soc. Am., 398, 143–171, doi:10.1130/2006.2398(09).

Attal, M., and J. Lavé (2009), Pebble abrasion during fluvial transport: Experimental results and implications for the evolution of the sediment load along rivers, J. Geophys. Res., 114, F04023, doi:10.1029/2009JF001328.

Brewer, P. A., and J. Lewin (1993), In-transport modification of alluvial sediment: Field evidence and laboratory experiments, Spec. Publ. Int. Assoc. Sedimentol., 17, 23–35.

Carretier S., Regard, V., Leanni, L. and Farias, M. (2019). Long-term dispersion of river

gravel in a canyon in the Atacama Desert, Central Andes, deduced from their 10Be concentrations. Scientific Reports, 9, 17763, https://doi.org/10.1038/s41598- 019-53806-x.

Dingle, E. H., Attal, M. & Sinclair, H. D. Abrasion-set limits on Himalayan gravel fux. Nature, 544, 471+, https://doi.org/10.1038/ nature22039 (2017).

Fedele, J. J. & Paola, C. Similarity solutions for fuvial sediment fining by selective deposition. J. Geophys. Res. Earth Surf. 112, F02038, https://doi.org/10.1029/2005JF000409 (2007).

Ferguson, R., T. Hoey, S. Wathen, and A. Werritty (1996), Field evidence for rapid downstream fining of river gravel through selective transport, Geology, 24(2), 179–182, doi:10.1130/0091-7613

Kuenen, P. H. (1956), Experimental erosion of pebbles: 2. Rolling by current, J. Geol., 64, 336–368, doi:10.1086/626370.

Lajeunesse, E., Devauchelle, O. & James, F. Advection and dispersion of bed load tracers. Earth Surf. Dyn. 6, 389–399, https://doi. org/10.5194/esurf-6-389-2018 (2018).

Nikora, V., Habersack, H., Huber, T. & McEwan, I. On bed particle diffusion in gravel bed flows under weak bed load transport. Wat. Resour. Res. 38, https://doi.org/10.1029/2001WR000513 (2002).

Paola, C. et al. Downstream fining by selective deposition in a laboratory fume. Sci. 258, 1757–1760 (1992).

Phillips, C. B. & Jerolmack, D. J. Dynamics and mechanics of bed-load tracer particles. Earth Surface Dynamics 2, 513–530, https:// doi.org/10.5194/esurf-2-513-2014 (2014).

Sklar, Leonard S. and Riebe, Clifford S. and Marshall, Jill A. and Genetti, Jennifer and Leclere, Shirin and Lukens, Claire L. and Merces, Viviane (2017). The problem of predicting the size distribution of sediment supplied by hillslopes to rivers, Geomorpholgy,

10.1016/j.geomorph.2016.05.005.

Whittaker, A. C. et al. Decoding downstream trends in stratigraphic grain size as a function of tectonic subsidence and sediment supply. Geol. Soc. Am. Bull. 123, 1363–1382, https://doi.org/10.1130/B30351.1 (2011).
* * *

---

## Author Response (AR1)

**ESD response to referees**

A.A. Sipos, G. Domokos, J. Török

January 10, 2021

**1 Response to Prof. Furbish**

We would like to thank Professor Furbish for a thorough and encouraging review. His comment about riverbed collisions was an eye-opener. Below we address all the issues raised in the report.

1. **APPLICATION OF THE MODEL TO ABRASION IN RIVERS**
   We propose to add a new subsection, after Subsection 2.4, where we discuss how our model can be explored to make predictions about fluvial environments. Here follows the suggested subsection:

**Fluvial abrasion**

**1.1 Fluvial abrasion**

Here we interpret the intuitive picture of fluvial abrasion in the context of our statistical model. In our model a fluvial environment may be represented by a *fluvial population*, consisting of $N + 1$ particles: a very large number ($N$) of small particles $X^i$ ($i = 1, 2, \ldots N$) representing the pebbles carried by the river and one very large particle $Y$ representing the riverbed. Such a scenario can not be explored directly in the context of our continuum model, however, as we will discuss in detail in Subsection 3.3, the discrete model can capture this situation even in the limit as $N \to \infty$.

To make a meaningful characterization of geologically relevant scenarios, we will regard two extreme cases which represent brackets on geological processes. In both cases we assume that the mass evolution is driven by binary collisions and we regard the limit as $N, Y \to \infty$ (while the masses $X^i$ of the small particles remain finite). Since we are interested in the mass evolution of pebbles (and in the current paper we are not interested in the mass evolution of the riverbed) we will denote the relative variance of the pebble population (i.e. all $X^i$ particles, the riverbed $Y$ not included) by $R(t)$. Our aim is to establish the sign of $R_t(t)$ as the main qualitative feature of collective dynamics.

In the first extreme scenario we assume that particles are chosen uniformly from the full *fluvial population*: i.e., the riverbed has no special role. In this case *almost all* collisions will happen among a pair of small particles $(X^i, X^j)$ thus the presence of the riverbed has no impact on the evolution of $R(t)$. For this extreme case all predictions of our continuum model remain valid: $r = 0.5$ will be a critical parameter value above which we see focusing $(R_t < 0)$, below which we see dispersing $(R_t > 0)$ behaviour. At the critical value $r = 0.5$ our model predicts neutral behaviour with $R_t = 0$.

In the second extreme scenario we assume that the small particles *exclusively* collide with the riverbed (large particle), i.e., we only have $(X^i, Y)$-type collisions. This means that the evolution for each of the small particles is an identical, independent two-particle process governed by the model (1)-(2) for binary collisional mass evolution. In this process, in the $Y \to \infty$ limit each individual small particle $X^i$ will thus evolve as

$$X^i(t) = X^i(0)e^{-t} \tag{11}$$

and thus follow Sternberg's Law. It is easy to show that for any initial distribution for the masses $X^i(0)$, in this process we have $R_t = 0$. The large $Y$-particle (riverbed) will lose some mass as well but in this publication we are not interested in that part of the process.

Intuitively it is clear that any geologically relevant process is in-between the above two extreme cases and, although we do not deliver a rigorous proof, it appears plausible that in a geologically relevant setting $R_t$ will be also bounded by the two evolutions predicted for the two extreme scenarios. As for the second extreme scenario we have $R_t = 0$ we expect that for any intermediate scenario the sign of $R_t$ will agree with the sign of $R_t$ based on the first extreme scenario. This would imply that all our qualitative predictions remain valid in fluvial environments.

2. **ESTIMATE FOR EXTREMAL $r$ VALUE**. We propose to add a new Appendix (Appendix D) where we explain our estimate for the parameter $r$ in case of smooth gradient flows (laminar flows). Here follows the suggested Appendix:

**Estimating physically possible values of $r$**

In the paper we assumed that the particle collision probability depends on the volume of the particles as

$$P(X) \propto X^r. \tag{D1}$$

Here we investigate two extreme scenarios, associated with the collision probabilities $P_{\text{smooth}}(X)$ and $P_{\text{turbulent}}(X)$ where we expect $r$ to assume its extreme values.

The first is the smooth gradient flow. In such a case the driving fluid has a strong, but on a particle-size scale constant velocity gradient in one of the spatial directions. Such situations may arise, e.g., in shallow water layers. Here the relative velocity of the particles grows with the distance. If we are at distance $u$ from the center of the particle in the direction of the flow velocity gradient, the collision probability $P_{\text{smooth}}(X)$ can be estimated by the product of the velocity difference and the linear cross-section of the particles. (Note that $R \equiv X^{1/3}$ is the linear size of the particle):

$$P_{\text{smooth}}(X) \sim \frac{1}{R} \int_0^R u\sqrt{R^2 - u^2}\,du = \frac{1}{3}R^2 = \frac{1}{3}X^{2/3}. \qquad \text{(D2)}$$

Based on (D1), this gives us an estimate for high $r = 2/3$.

The other extreme case is a fully chaotic motion where equipartition takes place [Uberoi, 1957]. Thus the kinetic energy of the particles ($\frac{1}{2}\rho X v^2$) is independent of their volume. Thus the speed of the particles must be proportional to $X^{-1/2}$. If we disregard correlations the particles have a cross-section proportional to their projected area which is proportional to $X^{2/3}$. Combining the two gives us

$$P_{\text{turbulent}}(X) \sim X^{-1/2}X^{2/3} = X^{1/6} \qquad \text{(D3)}$$

and based on (D1) we obtain $r = 1/6$. Thus it is possible to have physical scenarios apparent in nature where the value of $r$ falls to either side of the critical value of $r_c = 0.5$ with large enough margin.

3. **DIFFUSION**

The referee is right, we have used the term 'diffusion' incorrectly. We modified the text in Subsection 2.2., the new version is below.

Note that contrary to the majority of Fokker-Planck models, our model contains solely the advection term, which readily follows from the deterministic nature of the kernel. Here we aim to figure out the collective behavior implied by (7). Nonetheless, a stochastic kernel would produce diffusion in the master equation, such a generalization might be essential for testing model predictions against experimental data.

**2    Response to Dr. Bertoni**

While our paper is admittedly a theoretical model, still, its main goal is to offer a new toolkit to geologists. It is reassuring to know that this approach was vetted by an expert field geologist and we would like to sincerely thank Dr. Bertoni for his comments.

We very much appreciate his positive opinion on the problem statement and result discussion. This gives us hope that field experts may, in the future, engage in exploring this interesting subject.

Our referee raises the question how our theoretical predictions may be tested. While we regard this as a key issue, we did not elaborate on this aspect since no testing has been done so far. Still, some of the potential testing strategies appear to be quite clear as we discuss it below.

Our model can be tested at two levels:

(a) at the *input level*, i.e. by testing the kernel and

(b) at the *output level*, i.e. by testing pebble size distributions.

To perform (a), one would need time evolutions for full mass distributions. We are not aware of any such publicly available dataset. To obtain such data the best options appear to be either laboratory flume or drum experiments or, in the field, radio-tagged pebble experiments.

To perform (b), one would need size dependence of abrasion rates. Such an experiment is reported in Figure 9/a of [Attal and Lavé, 2009] and we fitted our kernel to their data.

We discuss these testing options in Subsection 1.4 of the modified manuscript and we included the (b)-type kernel fit in the new Appendix A. Here we found that the experimental data is in fair agreement with our kernel.

In addition we remark that in all experiments the validity depends on the ability of the experimenter to make measurements on the same particle population. To make the experiments consistent with this study, the most straightforward approach is to track the evolution R(t) of the relative variance.

(1) Tumbler experiments. In this case the validity of the experiment is automatically guaranteed. The energy level of the experiment may be controlled either by adjusting the speed or by adding water. Recording $R(t)$ at a wide range of energy levels may help to confirm some aspects of the presented theory.

(2) Flume experiments. Here again, the validity of the experiment is guaranteed. Circular flumes may be adequate testing platforms for lower energy levels.

(3) Field experiments. In this case both the validity and consistency of the experiment is a hard question. The most plausible option are radio-tagged particles, however, low recovery rates may prohibit a reliable monitoring of $R(t)$.

The referee also raises the question whether and to what extent our theory may help to distinguish between coastal and fluvial environments. The question is justified, yet, in the absence of experimental results, a full answer is lacking. Still, this question may be a main motivation behind the design of targeted experiments. Based alone on the predictions of the paper, focusing and dispersing behavior may be present both in a fluvial and in a coastal setting. Focusing processes operate at lower energy levels: this may characterize the lower reaches of rivers as well as wave-current-driven frictional abrasion in coastal environments.

These are the scenarios where our theory predicts that abrasion and transport act in a similar manner on mass distributions. On the other hand, high energy levels indicate dispersing processes where abrasion and transport are counteracting. Such scenarios may be observed in the upper reaches of rivers as well on high-energy beaches often visited by storms. As we can see, at higher energies both transport and abrasion operate much faster and it is a truly challenging question to find out which of these natural processes dominates. Our study offers a tool to make a meaningful statement: by measuring $R(t)$ in any of these settings, one can safely decide this question. We thank the referee for indicating minor points in the manuscript. In the resubmitted version, beyond the summary of the above ideas, we will correct those points. Once again, we sincerely thank for the report which raised fundamental questions and motivated us to think further about the applicability and testing of the proposed theory.

**3   Response to Dr. Carretier**

We thank our referee for a supportive review and also for raising some key issues, addressing which could substantially improve our manuscript.

Below we give a detailed, point-by-point response. also indicating changes in the resubmitted version.

*Referee' comment:*

*At the first order, the authors could increase the impact of their findings by better relying on and linking it more closely to geological and experimental observations. The authors claim that their model "fits" the geological observations, but I see no clear evidence of such a fit in their paper. Several assertions lack references (see specific comments), in particular concerning the relative role of selective transport and attrition in the evolution of grain size along a river. In fact, selective transport has been mainly advanced to explain or model downstream fining in rivers (e.g. Paola et al., 1992; Ferguson et al., 1996; Fedele and Paola, 2007; Whittaker et al., 2011 - see list of papers at the end of this review). In parallel, several authors have shown evidence of attrition to explain the variation in grain size (e.g. Brewer and Lewin, 1993; Attal and Lave, 2006; Dingle et al., 2017). These studies are worth mentioning and discussing to show that the results presented in this manuscript are consistent with geological observations.*

**Our response:** We agree with the referee that the list of references related to geological observations was far from complete in the original submission. In the modified manuscript we included the references suggested by the referee. We also agree with the referee that saying that our model "fits" geological observations was not a fortunate selection of expression. Certainly, as we pointed out in the main text, we did not fit the predictions of the model to any dataset - for the simple reason that currently no such dataset is available (which we discuss below). We modified the text by saying that "the model does not contradict existing geological observations."

*Referee' comment:*

*Theoretical results could also be confronted with experimental data, which are rare and valuable. In addition to Kuenen (1956), Attal and Lavé (2009) presented pebble attrition experiments on a 1:1 scale, representing Himalayan rivers conditions. These experiments show in particular that the mass loss by attrition increases with particle velocity but is weakly dependent on particle size. As the parameter r can represent the degree of dependence between the attrition rate and the particle size, a discussion on the possible value of r in the experiments of Attal and Lavé (2009) would allow to link the theory, these experiments and the prediction of the evolution of the relative variability (focusing or dispersing) in a natural case like that of Himalayan rivers.*

**Our response:**

Our model can be tested at two levels:

(a) at the *input level*, i.e. by testing the kernel and

(b) at the *output level*, i.e. by testing pebble size distributions.

To perform (a), one would need time evolutions for full mass distributions. We are not aware of any such publicly available dataset. To obtain such data the best options appear to be either laboratory flume or drum experiments or, in the field, radio-tagged pebble experiments.

To perform (b), one would need size dependence of abrasion rates. Such an experiment is reported in Figure 9/a of [Attal and Lavé, 2009] and we fitted our kernel to their data.

We discuss these testing options in Subsection 1.4 of the modified manuscript and we included the (b)-type kernel fit in a new Appendix (Appendix A). Here we found that the experimental data is in fair agreement with our kernel.

***Referee' comment:***

*Statistical physics is used here in a very clever way to describe the phenomenon of attrition. However, statistical physics (and the Fokker-Planck equation) has also been used to describe the transport of grains in a river. In this case, advection-diffusion emerges from a combination of probability densities that describe the distribution of the transport distances of the particles and that of their residence time at rest (e.g. Lajeunesse et al., 2018; Nikora et al., 2002). Longer residence times for larger particles (selective sorting) may explain some of the anomalous scattering observed in tracer data (Phillips and Jerolmack, 2014; Carretier et al., 2019). In a natural system that combines both the statistical physics of transport and abrasion, how could the proposed theory be verified on the basis of field observations? What experiments or observations need to be carried out to distinguish between these two components? Again, linking this proposed theory with predictions that could be made by geologists or experimenters would increase the scope of this paper. I recommend significantly increasing the discussion in this direction.*

**Our response:**

The relative effects of transport versus abrasion in downstream grain size evolution has been discussed in [Miller et al., 2014]. In the modified manuscript we included this reference. Direct comparison with our model could be performed, as discussed above, either at the input level (by comparing size dependence of abrasion rates) or by measuring complete time evolutions for pebble

size distributions. We mention these option in Subsection 1.4. of the modified manuscript.

*Referee' comment:*

*In the discussion, always with the aim of linking this theory with the field, I suggest discussing the following points: The pebbles impact each other but also the bedrock (which could represent a much larger volume for the population y). How would this change the result? The lateral contribution of the hillslopes modifies the initial distribution fo(x) in the real case (e.g. Attal and Lavé, 2006; Sklar et al., 2017). Can we find a natural case where the proposed theory could be tested (canyon with one particular lithology at some localized point upstream, providing tracer pebbles that could be followed downstream for example)?*

**Our response:** We included a new subsection (Subsection 2.5) in the modified manuscript where we discuss how our model can incorporate fluvial abrasion. We show that the actual fluvial process can be bracketed by two scenarios for which our model makes the same qualitative predictions, so we may regard these predictions as valid.

*Referee' comment:*

*The structure of this manuscript is confusing: the conclusions are described in several details in the introduction. This has the merit of setting reader's minds, but when reading it for the first time, I wondered whether it was a reminder of previous work or new findings. To guide the reader, I think it is important to write clearly what the addressed question is in the introduction.*

**Our response:**

We appreciate the referee's comment and we re-structured the Introduction: Subsection 1.2 discusses earlier work and Subsection 1.3 summarizes our model and our results.

*Specific comments*

*Page 2 Geological observations: extend this part based on above comments.*

Done.

*Page 2 Line 27-29: gives references.*

Added [Huber et al., 2020].

*Page 2 "as we can see": not obvious to me.*

Here we only refer to the fact that in some geological settings the dominance of transport has been observed, in other settings substantial contribution from abrasion has been observed. This has been stated above and "as we can see" refers to this ambiguity.

*P3 L14: "in (Domokos and Gibbons, 2013) " wrong citation format for this reference throughout the text: in Domokos and Gibbons (2013)*

We used identical citation format fro all references, also for this one. On the other hand, we noticed that the bibliographical data for this paper was outdated and we corrected that error.

*P4 Equation for Xt, Yt: number the equations. Give the numbers of these equations in the original paper and the corresponding notations, to help the reader.*

Done.

*P5 L9 Useful assumption to avoid dealing with the diffusion term in the F-P equation, but the experiments of Attal and Lavé (2009) show that fragmentation can be significant.*

We agree that in energetic environments fragmentation could be significant. We considered this effect in earlier papers [Szabó et al., 2013]. However, our model is only treating the chipping phase of abrasion. To capture fragmentation, the kernel would need to be expanded and that would certainly not admit any analytical treatment. In this paper our goal was to provide a model which is capable to approximate a range of physically relevant scenarios (not all) and which, on the other hand, admits efficient analytical approximations. Further work could certainly address fragmentation by the generalization of this model.

*P5 L21 "setting this is not the case": Why?*

In the collective setting the size refers to the independent variable of a distribution. As the distribution evolves, parameters of the distribution evolve in time (and can be differentiated with respect to time). However, the time variation of the size $x$ is not interpreteted in this setting.

*P5 L31 "that the that the"*

Done.

*P6 L10: Is the correspondence with Sternberg's law demonstrated in this paper or in a previous paper?*

It is demonstrated in [Domokos and Gibbons, 2018]. However, in our paper the plots in the first row of Figure 3 also show that the expected value evolves according to Sternberg's Law. This is also shown analytically in Appendix C1 for the truncated kernel.

*P7 L6-10. I think it would be useful to discuss the meaning of the two populations. Are they particles of the bedload? Of the suspended load? Both? Does the collision model reflect a dynamic where the grains are in suspension? In saltation? I imagine that the physics of collisions is different in these cases.*

In the modified manuscript we added Subsection 2.5 on fluvial environments where one possible geophysical interpretation of outliers is provided (the riverbed may be regarded as one single huge outlier). This does not exclude other interpretations, though.

*P8 L7. Could you give a "hands-on" explanation of the nature of diffusion related to fragmentation in this case?*

This line was misleading. Our model does not include diffusion but the absence of diffusion is not equivalent to prohibiting fragmentation. We altered the

text in the paper accordingly.

*P8 L25 Summary or summation ?*
Summation. Corrected.

*P8 L26-30 Explain why you are testing these other kernels whereas you announce in the introduction that the kernel finally used is "the right one".*
We explained this above. We investigate the simple kernels just to show that one can NOT get away with them: while they admit analytical treatment they do not meet the most fundamental geophysical requirements. Since this is the first paper aiming to establish a statistical theory for mass evolution, we thought it would be appropriate to indicate why we do not use any of the simple kernels. On the other hand, more complicated kernels may be closer to the physics of the process but would be, most likely, beyond the reach of analysis.
We describe that our choice for the kernel is a trade-off between physical accuracy and mathematical transparency.

*P9 L12: Proportional TO projected ? (could you give a reference?)*

Corrected. Appendix D added.

*P9 L21-27 Give references or explain better the values deduced for r. Again, do these r values depend on the type of transport (suspension, saltation etc.)?*
The estimates on the parameter $r$ are now better explained in Appendix D.

*P9 L 26-27 Important to make the link with the field. Expand on this aspect.*
We added Appendix D to explain the values of $r$. On the other hand, making specific predictions for field data appears to be a bit far-fetched at this stage. We do believe that $r$ is characteristic of the environment. The fitting of the kernel to the data of Attal and Lavé in Appendix A shows that energetic fluvial environments correspond to low values of $r$, as we indicated. We do not have other data currently.

*P9 L32 what is V? (Volume I think but I have not seen its definition above).*
Corrected. It should have been $X$ (not $V$).

*P11 L10 justify why you take a log-normal law for fo. Does the result change if you take another law?*
We took a lognormal distribution because most mass distributions can be well approximated by this. None of the qualitative features of the model are influenced by the particular type of the distribution.

*P11 L11: "The applied time step is fixed at $\Delta t = 0.01$." Already said above.*
Corrected (removed).

*P11 Figure 4: Indicate c: continuous and d: discrete in the caption. Because the by-product of abrasion is removed, I would have expected that the distribution tends toward a Dirac at x=0 in all the cases. Why is it not the case? ("hands-on" explanation please;). In the focusing case, what determines the final x value of the Dirac?*

"c" and "d" added to caption.

The referee is right, in the focusing case the distribution converges onto the x=0 Dirac Delta. In our plots we followed the evolution only for finite time (t=5) so we see a very sharp peak in the vicinity of 0.

*P12-12 3.3 Very interesting discussion. In Figure 4 it could also be argued that the tail of the distribution between 2E-7 and 2E-6 is a power law, while the outliers represent a subsampling of the actual distribution, as often observed in seismology (for magnitude) or hydrology (for peak discharge). Could the persistence of these outliers indicate some fractional derivative component not taken into account in the Fokker Planck equation?*

A very interesting point. The Fokker-Planck equation in eq. (7) in the paper indeed resembles a problem with a spatial fractional derivative. It might be argued that a kernel could be selected such way that one of the widely used fractional derivatives formulas (such as the Caputo fractional derivative) is retrieved. However, to the best of our knowledge, the compound kernel in the paper does not establish any known fractional derivative formula. But this does not contradict the observation of the reviewer: we think that although the outliers are identified in finite samples of $N$ particles, the result in Appendix E is based on the $N \to \infty$ limit. This implies that a well-chosen bimodal distribution (with a small peak representing the outliers) would also predict the survival of the outliers, and this is also suggested in Figure 7. We think that although our model formally does not contain a fractional derivative, the behaviour of the tail of the distribution is very similar to models with an established fractional derivative.

*P14 "(a) existing geological observations": not demonstrated in my opinion.*

We added Subsection 1.4 about the two possible levels of testing our model and demonstrated in Appendix A that the model exhibits reasonable agreement with experimental data.

**References**

[revised manuscript text omitted]